# The interferon-rich skin environment regulates Langerhans cell ADAM17 to promote photosensitivity in lupus

Thomas Morgan Li[1,§§,‡], Victoria Zyulina[1,2,§§,‡], Ethan S Seltzer[1,§§], Marija Dacic[3,4,5], Yurii Chinenov[3], Andrea R Daamen[6], Keila R Veiga[1,7,8,#], Noa Schwartz[1,9,¶], David J Oliver[3], Pamela Cabahug-Zuckerman[1], Jose Lora[4,5], Yong Liu[10], William D Shipman[1,11,12,**], William G Ambler[1,7,8], Sarah F Taber[7,8], Karen B Onel[7,8], Jonathan H Zippin[10], Mehdi Rashighi[13], James G Krueger[14], Niroshana Anandasabapathy[10,11,12], Inez Rogatsky[2,3,4,12], Ali Jabbari[14,††], Carl P Blobel[4,5,15], Peter E Lipsky[6], Theresa T Lu[1,2,7,8,12,*]

*For correspondence: lut@hss.edu

Present address: ‡Icahn School of Medicine at Mount Sinai, New York, United States; §Graduate Program in Bioscience, Rockefeller University, New York, United States; #Department of Rheumatology, Maria Fareri Children's Hospital, New York, United States; ¶Department of Medicine (Rheumatology), Montefiore Medical Center/Albert, Einstein College of Medicine, New York, United States; **Department of Dermatology, Yale School of Medicine, New Haven, United States; ††Department of Dermatology, University of Iowa Hospital and Clinics, Iowa, United States

§§These authors share first authorship

¹Autoimmunity and Inflammation Program, Hospital for Special Surgery Research Institute, New York, United States; ²Department of Microbiology and Immunology, Weill Cornell Medical College, New York, United States; ³David Z. Rosensweig Genomics Research Center, Hospital for Special Surgery Research Institute, New York, United States; ⁴Arthritis and Tissue Degeneration Program, Hospital for Special Surgery Research Institute, New York, United States; ⁵Physiology, Biophysics, and Systems Biology Program, Weill Cornell Graduate School of Medical Sciences, New York, United States; ⁶Department of Medicine, AMPEL BioSolutions, Charlottesville, United States; ⁷Pediatric Rheumatology, Department of Medicine, Hospital for Special Surgery, New York, United States; ⁸Department of Pediatrics, Weill Cornell Medical College, New York, United States; ⁹Rheumatology, Department of Medicine, Hospital for Special Surgery, New York, United States; ¹⁰Department of Dermatology, Weill Cornell Medical College, New York, United States; ¹¹Weill Cornell/Rockefeller/Sloan-Kettering Tri-Institutional MD-PhD Program, Weill Cornell Medical College, New York, United States; ¹²Immunology and Microbial Pathogenesis Program, Weill Cornell Graduate School of Medical Sciences, New York, United States; ¹³Department of Dermatology, University of Massachusetts Medical School, Worcester, United States; ¹⁴Laboratory of Investigative Dermatology, Rockefeller University, New York, United States; ¹⁵Department of Physiology, Biophysics, and Systems Biology, Weill Cornell Medical College, New York, United States

**Abstract** The autoimmune disease lupus erythematosus (lupus) is characterized by photosensitivity, where even ambient ultraviolet radiation (UVR) exposure can lead to development of inflammatory skin lesions. We have previously shown that Langerhans cells (LCs) limit keratinocyte apoptosis and photosensitivity via a disintegrin and metalloprotease 17 (ADAM17)-mediated release of epidermal growth factor receptor (EGFR) ligands and that LC ADAM17 sheddase activity is reduced in lupus. Here, we sought to understand how the lupus skin environment contributes to LC ADAM17 dysfunction and, in the process, differentiate between effects on LC ADAM17 sheddase function, LC ADAM17 expression, and LC numbers. We show through transcriptomic analysis a shared IFN-rich environment in non-lesional skin across human lupus and three murine models: MRL/lpr, B6.Sle1yaa, and imiquimod (IMQ) mice. IFN-I inhibits LC ADAM17 sheddase activity in murine and human LCs, and IFNAR blockade in lupus model mice restores LC ADAM17 sheddase activity,

all without consistent effects on LC ADAM17 protein expression or LC numbers. Anti-IFNAR-mediated LC ADAM17 sheddase function restoration is associated with reduced photosensitive responses that are dependent on EGFR signaling and LC ADAM17. Reactive oxygen species (ROS) is a known mediator of ADAM17 activity; we show that UVR-induced LC ROS production is reduced in lupus model mice, restored by anti-IFNAR, and is cytoplasmic in origin. Our findings suggest that IFN-I promotes photosensitivity at least in part by inhibiting UVR-induced LC ADAM17 sheddase function and raise the possibility that anifrolumab ameliorates lupus skin disease in part by restoring this function. This work provides insight into IFN-I-mediated disease mechanisms, LC regulation, and a potential mechanism of action for anifrolumab in lupus.

## Editor's evaluation

This study presents a useful assessment of the possible role of type I interferons in inhibiting Adam17 protease/sheddase activity and their correlation with decreased Langerhans Cells signature in lesional and nonlesional CLE and murine models as cause of photosensitive lupus. The data were collected and analyzed using solid methodology. This work will be of interest to scientists interested in photosensitivity in the setting of lupus.

## Introduction

The autoimmune disease lupus erythematosus (lupus) manifests in part by photosensitivity, a sensitivity to ultraviolet radiation (UVR) whereby even ambient sunlight can trigger the development of inflammatory skin lesions (*Foering et al., 2013*; *Kim and Chong, 2013*). In patients with the systemic form of the disease (SLE), with or without diagnosed cutaneous lupus (CLE), photosensitive skin responses can also be associated with worsening systemic autoimmunity and end organ damage. Photosensitive skin inflammation, the accompanying risk of systemic disease flares, and the lifestyle modifications such as sun avoidance needed to prevent UVR-induced skin inflammation all contribute to reduced quality of life (*Bachen et al., 2009*; *Foering et al., 2012*). Insight into mechanisms that underlie photosensitivity remain limited (*Estadt et al., 2022*; *Sim et al., 2021*), and better understanding of pathophysiology will help to advance therapeutic options.

To study the pathophysiology that drives photosensitivity, we have focused on studying non-lesional skin to delineate mechanisms that have already gone awry that lead to photosensitivity. We recently reported that Langerhans cells (LCs) can limit UVR-induced skin inflammation by expressing a disintegrin and metalloprotease 17 (ADAM17), which releases membrane-bound epidermal growth factor receptor (EGFR) ligands that then preserve epidermal integrity via EGFR stimulation of keratinocytes (*Shipman et al., 2018*). Two distinct photosensitive SLE mouse models showed a reduction in LC *Adam17* mRNA in non-lesional skin which corresponded to reduced LC ADAM17 sheddase activity, and we could reduce photosensitivity by circumventing the reduced EGFR ligand shedding with topical EGFR ligand application. Together, these data suggested that reduction of LC ADAM17-dependent EGFR-ligand sheddase activity contributes to photosensitivity. However, no murine ADAM17 antibody was available to differentiate between diminished LC ADAM17 protein level versus sheddase function. Non-lesional skin of SLE patients showed reduced epidermal EGFR phosphorylation and a subset had reduced LC numbers, suggesting that reduced LC ADAM17 sheddase function, LC ADAM17 expression and/or LC numbers could potentially contribute to pathology. Thus, our previous study put forth a new model for LC dysfunction in lupus photosensitivity but the mechanisms that lead to LC dysfunction and distinguishing among the regulation of LC ADAM17 sheddase function, protein expression, and LC numbers remained to be fully elucidated.

A prominent interferon (IFN) signature indicative of exposure to type I interferon (IFN-I) is found in tissues and circulating cells in lupus (*Baechler et al., 2003*; *Bennett et al., 2003*; *Catalina et al., 2019*; *Der et al., 2019*; *Jabbari et al., 2014*; *Kirou et al., 2004*; *Martínez et al., 2022*; *Psarras et al., 2020*; *Psarras et al., 2022*), and the recent FDA approval of anifrolumab (anti-IFNAR1) for SLE highlights the importance of IFN-I in disease pathogenesis. Numerous studies have pointed to an IFN-I-rich environment in even non-lesional skin based on transcriptomic signatures of skin from SLE and CLE patients (*Billi et al., 2022*; *Der et al., 2017*; *Der et al., 2019*; *Psarras et al., 2020*), upregulation of keratinocyte IFN-κ expression in keratinocytes of CLE, SLE, and ANA +patients (*Psarras*

*et al., 2020*; *Stannard et al., 2017*), and upregulation of IFN-stimulated genes (ISGs) such as *MX1* on tissue sections in SLE, incomplete SLE, and ANA +patients (*Lambers et al., 2019*; *Psarras et al., 2020*; *Reefman et al., 2008*). Remarkably, the IFN signature in both lesional and non-lesional skin is greatly enriched compared to the signature in blood (*Psarras et al., 2020*), further pointing to a potential pathogenic role for IFN-I in lupus skin disease, and the IFN-I-rich environment in grossly unaffected skin points to a role for IFN-I in potentially predisposing to lesion development. Functionally, the reduced skin lesions in IFNAR-deficient SLE model mice (*Nickerson et al., 2013*) is consistent with this idea, and, indeed, anifrolumab is especially efficacious for skin disease in SLE patients (*Furie et al., 2017*; *Merrill et al., 2018*; *Morand et al., 2020*). Mechanisms by which IFN-I contributes to and anifrolumab exerts this ameliorative effect on lupus skin disease, however, remains to be better understood.

Here, we consider LC dysfunction in the context of the high IFN levels in lupus, and test the hypothesis that the IFN-I-rich environment present in even non-lesional lupus skin promotes LC dysfunction to contribute to photosensitivity. We extend accumulating human data showing an IFN-I signature in non-lesional CLE skin and establish that non-lesional skin from photosensitive MRL/lpr, B6.Sle1yaa, and imiquimod (IMQ) SLE models share IFN signatures with human skin. IFN-I was sufficient to inhibit LC ADAM17 sheddase activity without consistent effects on surface expression levels or altering LC numbers in healthy murine and human LCs. Conversely, anti-IFNAR treatment of SLE mouse models restored LC ADAM17 sheddase function without affecting expression levels or LC numbers. Anti-IFNAR restoration of LC ADAM17 sheddase function was associated with reduced photosensitive response, and this anti-IFNAR effect was dependent on EGFR signaling and LC ADAM17. Lastly, we show that UVR-induced reactive oxygen species (ROS) known to promote ADAM17 sheddase activity (*Singh et al., 2009*), was reduced in lupus model LCs, that anti-IFNAR restored ROS expression, and that ROS was of cytoplasmic origin. Together, our results establish that multiple photosensitive murine SLE models are similar to human lupus patients in demonstrating IFN signatures in non-lesional skin and put forth a mechanism whereby IFN-I contributes to photosensitivity at least in part by inhibiting LC sheddase ADAM17 activity. These data provide insight into IFN-I-mediated contributions to photosensitivity, delineate a driver of LC dysfunction, and suggest the possibility that restoration of LC ADAM17 function is a mechanism of action for anifrolumab in lupus skin disease.

## Results

### Non-lesional skin from human lupus and murine models share an IFN-rich environment

To understand mechanisms that lead to the LC defects seen in non-lesional skin of SLE mouse models and human SLE (*Shipman et al., 2018*), we assessed the non-lesional skin transcriptomic profiles from human lupus patients and multiple murine models. We examined a non-lesional skin microarray dataset from a CLE cohort with the discoid form of CLE (DLE) that had not been previously analyzed and was part of a cohort whose lesional skin was shown to have a Th1 signature when compared to healthy controls and psoriatic skin (*Jabbari et al., 2014*). The gene expression profile of the non-lesional DLE skin was significantly different from that of healthy skin (*Supplementary file 1*), lesional skin, and psoriatic skin (*Figure 1A*). Differential pathway analysis via Quantitative Set Analysis for Gene Expression (QuSAGE) showed that IFN, IFN-I and IFN-γ pathways were among the upregulated pathways in non-lesional DLE skin compared to normal controls (*Figure 1B* and *Supplementary file 2*). The IFN-I pathway was more prominent than the IFNg pathway, with upregulation of a number of IFN-I-associated transcription factors such as *IRF1, 6, 7, 8* and IFN-stimulated genes including *MX1, MX2, XAF1, IFI27*, and *ISG15* in non-lesional skin (*Figure 1C–D*, *Figure 1—figure supplement 1*). Comparison of non-lesional skin with lesional skin showed that this IFN response was overall less dramatic in non-lesional skin (*Figure 1D*, *Figure 1—figure supplement 1*), consistent with findings of previous studies (*Billi et al., 2022*; *Martínez et al., 2022*; *Psarras et al., 2020*). The more muted response did not apply to every gene in the pathway, however, with *ADAR, IRF6*, and *PTPN1* expressed at higher levels in non-lesional skin (*Figure 1D*, *Figure 1—figure supplement 1*) and suggesting a distinct biology in non-lesional and lesional skin. These data are consistent with that from other cohorts (*Billi et al., 2022*; *Der et al., 2017*; *Der et al., 2019*; *Psarras et al., 2020*) showing that non-lesional skin, similar to lesional skin, has an IFN-rich environment.

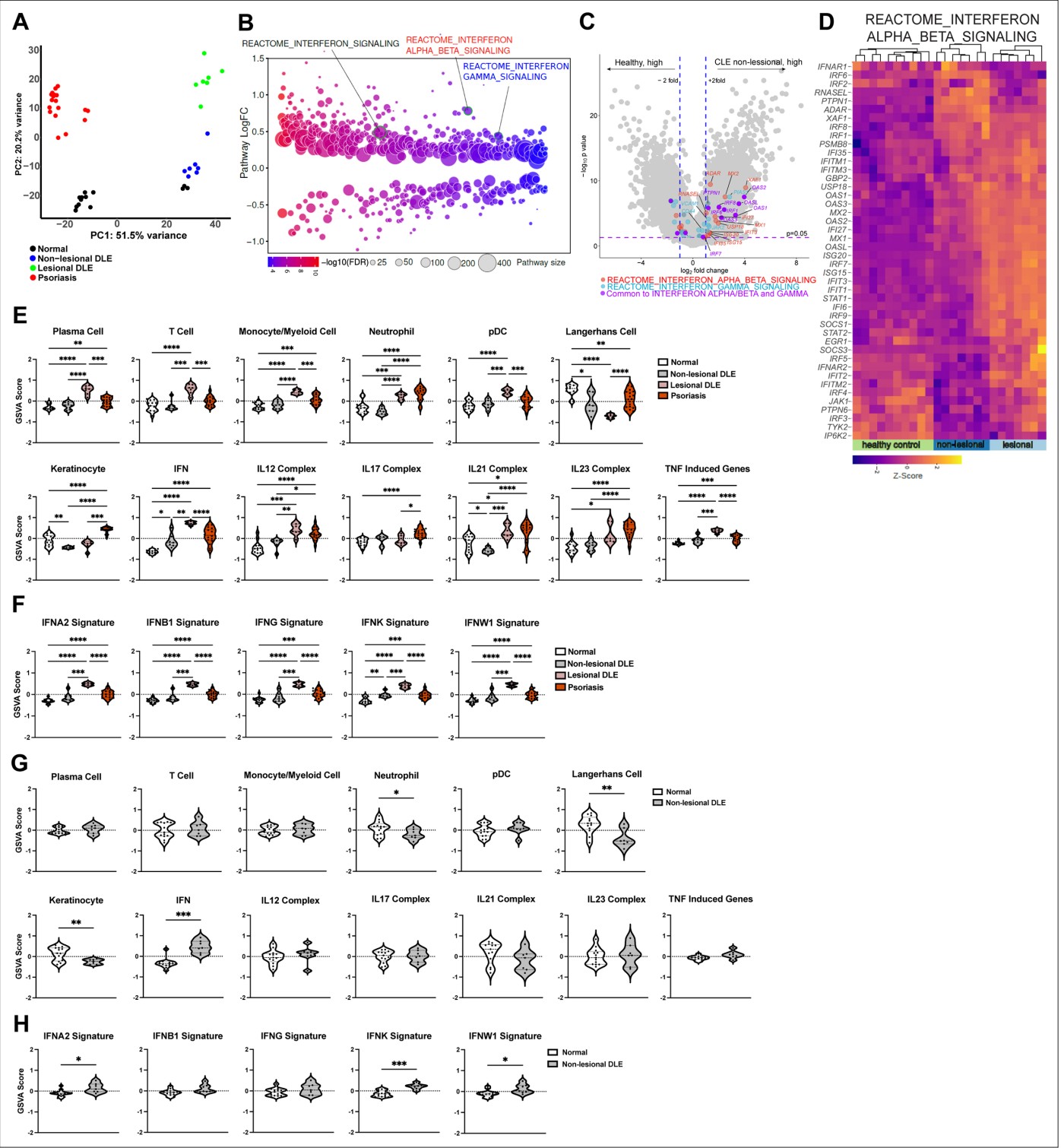

**Figure 1.** Analysis of a new cohort shows an IFN-rich environment in non-lesional DLE skin. (A–H) Microarray analysis of gene expression from non-lesional skin of DLE (n=7, this manuscript), lesional DLE (n=7), psoriasis patients (n=17), and healthy controls (n=13) (*Jabbari et al., 2014*). In (A, D–F), analyses include lesional DLE or both lesional and psoriasis samples. (A) Principal component analysis (PCA) of patient samples using top 500 genes. (B) Differentially expressed pathways in control and non-lesional DLE skin were determined using QuSAGE pathway analysis against Molecular Signatures Database (MSigDB). (C) Volcano plot of differentially expressed genes. Genes from IFN-α/β (red), IFN-γ (blue) pathways, and IRF transcription factors (green) are marked. (D) Heatmap of z-score transformed gene expression in the IFN-α/β signaling pathway. (E–H) Gene Set Variation Analysis

*Figure 1 continued on next page*

*Figure 1 continued*

(GSVA) of gene sets relevant to lupus (*Martínez et al., 2022*), with (**G–H**) comparing only control and non-lesional skin. (**F, H**) GSVA using gene sets comprising specific IFN subtypes. (**E–H**) *p<0.05, **p<0.01, ***p<0.001, ****p<0.0001 by unpaired t-test.

The online version of this article includes the following figure supplement(s) for figure 1:

**Figure supplement 1.** Further analysis of gene expression in human DLE.

We also analyzed the DLE data using Gene Set Variation Analysis (GSVA) to examine expression of defined lupus-relevant gene sets. We had previously used this approach to analyze publicly available bulk transcriptomic data from non-lesional and lesional CLE skin that included the DLE lesional skin data (*Martínez et al., 2022*). Reanalysis of the lesional skin here reiterated the GSVA results of multiple datasets (*Martínez et al., 2022*), showing enrichment of immune cell gene sets including plasma cells, T cells, monocyte/myeloid cells, neutrophils, and plasmacytoid dendritic cells (pDCs) when compared to controls (*Figure 1E*; *Supplementary file 3*). The LC gene set was de-enriched in lesional skin, echoing the LC loss seen in other studies (*Billi et al., 2022*; *Bos et al., 1986*; *Sontheimer and Bergstresser, 1982*), and this effect was more dramatic than the LC de-enrichment in lesional psoriatic skin (*Figure 1E*). Lesional DLE samples were also enriched for inflammatory cytokine gene sets, including IL12, IL21, IL23, TNF, and a highly upregulated IFN gene signature as previously seen (*Figure 1E*; *Martínez et al., 2022*). The upregulation of IFN response genes in lesional DLE skin extended to gene signatures indicative of response to specific IFNs, including IFN-Is (IFN-α, β, κ, and ω) and type II IFNs (IFN-γ; *Figure 1F*). The IFN-γ signature is consistent with the Th1 signature identified in the original study examining these gene expression data along with cellular phenotyping (*Jabbari et al., 2014*).

Non-lesional skin also showed upregulation of the IFN gene set in GSVA when compared to healthy controls, although this change was more modest than in lesional skin (*Figure 1E–G*), similar to that seen in the QuSAGE analysis (*Figure 1D*, *Figure 1—figure supplement 1*). IFN gene set changes in non-lesional skin were driven by IFN-α, IFN-κ, and IFN-ω gene set upregulation (*Figure 1H*). Non-lesional skin showed de-enrichment of the LC gene set and this effect was more modest than in lesional skin (*Figure 1E and G*), consistent with the noticeable loss of LCs in lesional skin compared to adjacent non-lesional skin by immunostaining of tissue sections (*Sontheimer and Bergstresser, 1982*). In contrast, non-lesional skin showed de-enrichment of the keratinocyte gene set, an effect not seen in lesional skin (*Figure 1E and G*), suggesting a distinct biology in non-lesional and lesional skin. GSVA, then, showed that while lesional skin was characterized by a proinflammatory IFN-rich environment and reduction of LC signals, non-lesional skin had a less inflammatory environment but one that was notable for an IFN response and LC reduction.

We next examined non-lesional ear skin from MRL/lpr, B6.Sle1yaa, and imiquimod (IMQ) SLE mouse models by RNAseq. The photosensitive MRL/lpr model has a *Fas* gene mutation and develops a lupus-like phenotype especially well on the MRL genetic background (*Menke et al., 2008*; *Theofilopoulos and Dixon, 1985*). MRL/lpr mice have reduced LC ADAM17 sheddase function and LC *Adam17* mRNA, and topical EGFR ligand supplementation ameliorated photosensitivity (*Shipman et al., 2018*). Although the differences in gene expression between MRL/lpr mice and control MRL-MpJ (MRL/+) mice by RNAseq involved fewer genes than those in the human cohort (*Figures 1A–D, and 2A–D*; *Supplementary file 4*), QuSAGE pathway analysis showed that the IFN-α/β and IFN-γ pathways were among the most highly expressed in MRL/lpr mice compared to control MRL/+mice (*Figure 2B* and *Supplementary file 5*). Consistent with the activation of IFN pathways, *IRF 1, 2, 7,* and *9* transcription factors and a number of their targets were expressed at higher levels in MRL/lpr mice (*Figure 2C–D* and *Figure 2—figure supplement 1A*).

The B6.Sle1yaa lupus model is driven by the *Sle1* lupus susceptibility locus derived from lupus-prone NZB2410 mice in combination with the Y chromosome autoimmune accelerator locus whose effects are attributed to *Tlr7* duplication (*Pisitkun et al., 2006*; *Subramanian et al., 2006*). B6.Sle1yaa mice are photosensitive and have reduced LC *Adam17* mRNA expression (*Shipman et al., 2018*). As expected, compared to B6 controls, B6.Sle1yaa mice expressed elevated levels of several lupus related genes from the *Yaa* locus including *Tlr7* (*Figure 2—figure supplement 1C*). Although differences between the B6 and B6.Sle1yaa transcriptomes were modest (*Figure 2E*; *Supplementary file 6*), several genes involved in the regulation of and response to IFN were expressed at higher levels, including *Irf7, Irf9, Isg15, Xaf1, Selp,* and *Ube2l6* (*Figure 2—figure supplement 1D*). LC numbers as

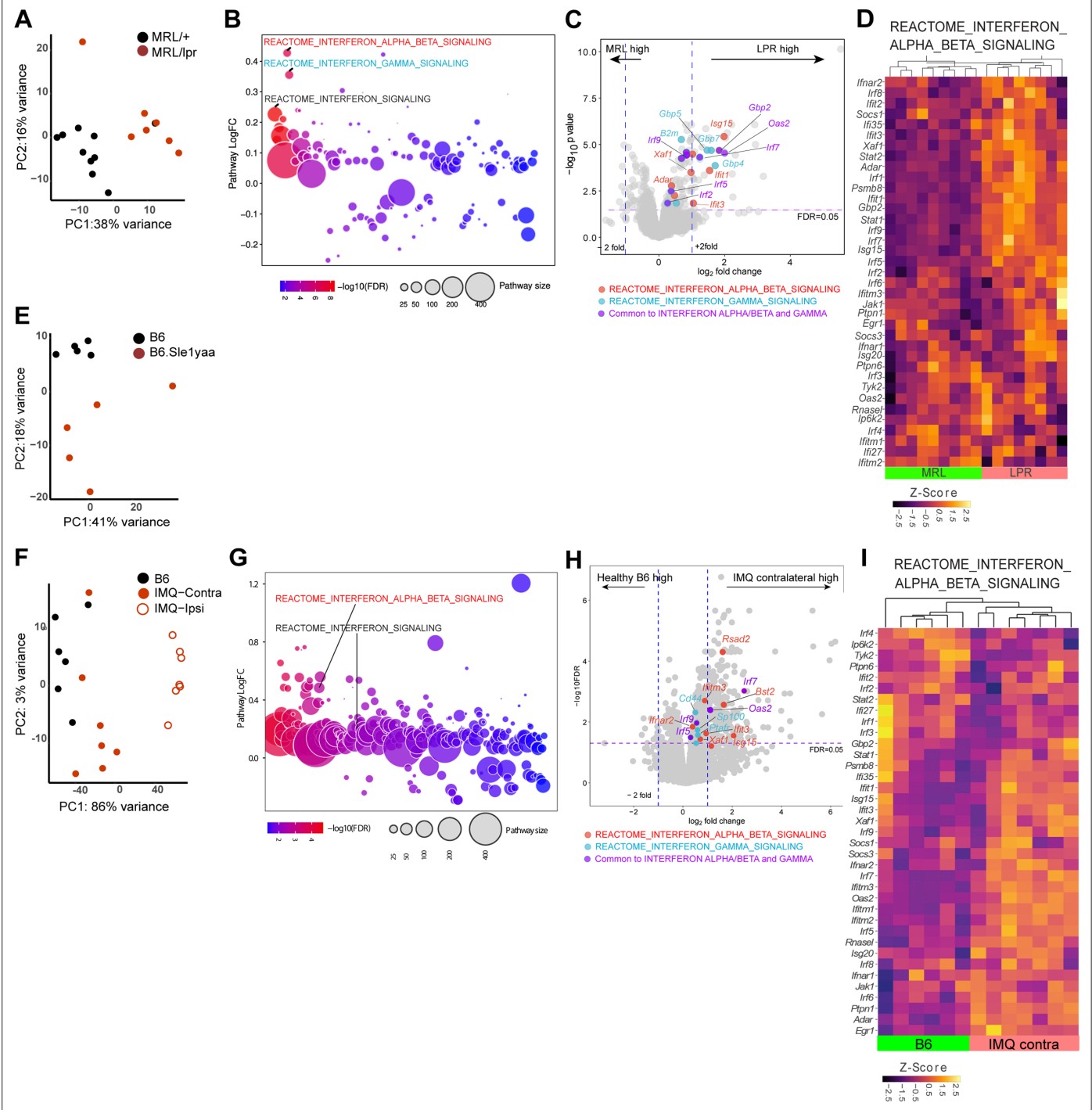

**Figure 2.** Three photosensitive SLE models show upregulated IFN-related gene expression in non-lesional skin by QuSAGE analysis. (**A–I**) RNAseq analysis of gene expression and pathway analyses from MRL/lpr (LPR) and control MRL/+ (MRL) (**A–D**), B6.Sle1yaa and control B6 mice (**E**), and IMQ and control B6 mice (**F–I**). For the IMQ model, mice were painted on one ear (ipsilateral) and the unpainted ear (contralateral) was taken as the non-lesional ear. (**A, E, F**) PCA using top 500 genes. (**B, G**) Differentially expressed pathways determined by QuSAGE pathway analysis against MSigDB. (**C, H**) Volcano plot of differentially expressed genes. Genes from IFN-α/β (red), IFN-γ (blue), and IRF transcription factor (green) pathways are marked. (**D, I**) Heatmap of z-score transformed gene expression in the IFN α/β signaling pathway. (**A, E, F**) Each symbol represents one mouse.

The online version of this article includes the following figure supplement(s) for figure 2:

**Figure supplement 1.** Further analysis of gene expression in skin of multiple murine lupus models.

a proportion of total cells were not different at 10 months of age when this model is fully diseased (*Shipman et al., 2018*), but *Cd207* expression was downregulated in these younger (5 month old) diseased mice when compared to B6 controls (*Figure 2—figure supplement 1D*), potentially hinting at either reduced LC numbers at this time point or altered CD207 expression on LCs.

The IMQ model is induced by 4–5 weeks of topical IMQ, a TLR 7/8 agonist, resulting in autoantibody production, enlarged spleen and lymph nodes, nephritis, and photosensitivity characteristic of SLE (*Ambler et al., 2022*; *Goel et al., 2020*; *Yokogawa et al., 2014*). The model can be induced in B6 mice, allowing for the use of transgenic models to assess for effects on lupus features without crossing onto a different genetic background. We applied IMQ to only one ear of B6 mice and assessed gene expression by RNAseq in both the painted ipsilateral ear and the unpainted contralateral 'non-lesional' ear. Similar to non-lesional skin in the MRL/lpr mice, the contralateral ear in IMQ mice demonstrated an IFN-I signature, with upregulation of a number of ISGs and *IRF 7* and *9* (*Figure 2F–I*, *Figure 2— figure supplement 1B*, *Supplementary file 7*, *Supplementary file 8*). In contrast to MRL/lpr mice, IFN gamma pathways were not upregulated in IMQ mice. By QuSAGE analysis, then, the non-lesional skin of MLR/lpr, B6.Sle1yaa, and IMQ SLE models showed similarities to human non-lesional skin in demonstrating an IFN-rich environment.

By GSVA using lupus-relevant murine gene sets (*Supplementary file 9*), MRL/lpr mice showed increased enrichment of a number of inflammatory cell and cytokine gene sets including T cells, myeloid cells, neutrophils, pDCs, IFN, IL1, IL21, and TNF (*Figure 3A*). The keratinocyte signature was enriched in MRL/lpr mice, in contrast to the de-enrichment of this gene set in human DLE (*Figure 1G*, *Figure 3A*). The LC signature did not show any quantitative changes; assuming that this gene set could in part reflect LC numbers, this result was consistent with unchanged LC numbers in the skin of MRL/lpr mice (*Shipman et al., 2018*). Similar to human DLE, the IFN gene set upregulation in MRL/lpr non-lesional skin was broad, with significant enrichment of gene signatures induced by stimulation from multiple IFNs (*Figure 3B*). B6.Sle1yaa mice showed less dramatic differences than the MRL/lpr model, but also showed upregulation of the IFN gene set without a particular contribution from any one IFN (*Figure 3C–D*). The LC gene set was de-enriched in B6.Sle1yaa mice (*Figure 3C*), driven in part by the reduction in *Cd207* (*Figure 2—figure supplement 1C*), potentially reflecting decreased LC numbers or altered LC phenotypes. The contralateral 'non-lesional' ears from IMQ-treated mice also exhibited significant upregulation of the IFN gene signature and were similar to MRL/lpr mice in showing enriched gene signatures of myeloid cells, IL21, and TNF (*Figure 3E*). T cell genes were uniquely de-enriched in IMQ skin (*Figure 3E*). The LC gene set was not altered (*Figure 3E*), although there was de-enrichment in IMQ-painted ipsilateral skin (*Figure 3—figure supplement 1*). Similar to human DLE and MRL/lpr mice, the IFN signature in non-lesional IMQ skin reflected enrichment of multiple IFN-I gene sets (*Figure 3F*). Interestingly, the overall IFN gene set was not enriched in the IMQ-painted ipsilateral skin (*Figure 3—figure supplement 1*), although there was enrichment in two of the specific IFN signatures (IFNB1 and IFNW1; *Figure 3—figure supplement 1*). This suggested a distinct biology such as tissue damage in the skin that received repeated direct exposure to IMQ. Together, the gene expression analysis revealed unique transcriptomic profiles across multiple murine SLE models and pointed to a shared IFN signature across non-lesional skin of all models and human DLE.

We assessed for common upregulated genes in human DLE, MRL/lpr, and IMQ datasets. While a direct compassion was complicated by the difference in data acquisition methods (microarray versus RNAseq), species and the magnitude of response between mouse models and human patients, utilizing a common size effect (logFC = 0.5) and FDR cutoffs (<0.05) yielded a small group of upregulated genes common to all three datasets (*Figure 4A*). All of these genes (*Sprr1b, Isg15, Ddx60, Bst2, Xaf1, Rsad2, Slc7a8, Ifit3*) were interferon-inducible genes. Similarly, in pairwise set overlaps (DLE – MRL/lpr, MRL/lpr – IMQ or IMQ – MRL/lpr) large proportion of overlapping genes were IFN-inducible targets including Ifi genes (*Ifi44, 203, 204*), IRF transcription factors (*Irf 1,2,7, 8,* and *9*), *Oas 2,3* and *L2, Mx1* (*Supplementary file 10*).

Comparisons of the GSVA analyses of non-lesional skin among the datasets (*Figure 4B*) also emphasized the prominent enrichment of IFN genes in human DLE and MRL/lpr, B6.Sle1yaa, and B6 IMQ SLE models. Notably, IFN upregulation correlated with loss of LCs in human DLE and B6.Sle1yaa mice but not in MRL/lpr and IMQ mice (*Figure 4B*). The lack of correlation in IMQ mice also extended to the painted ipsilateral ear, where there was a reduced LC signature in the absence of IFN upregulation

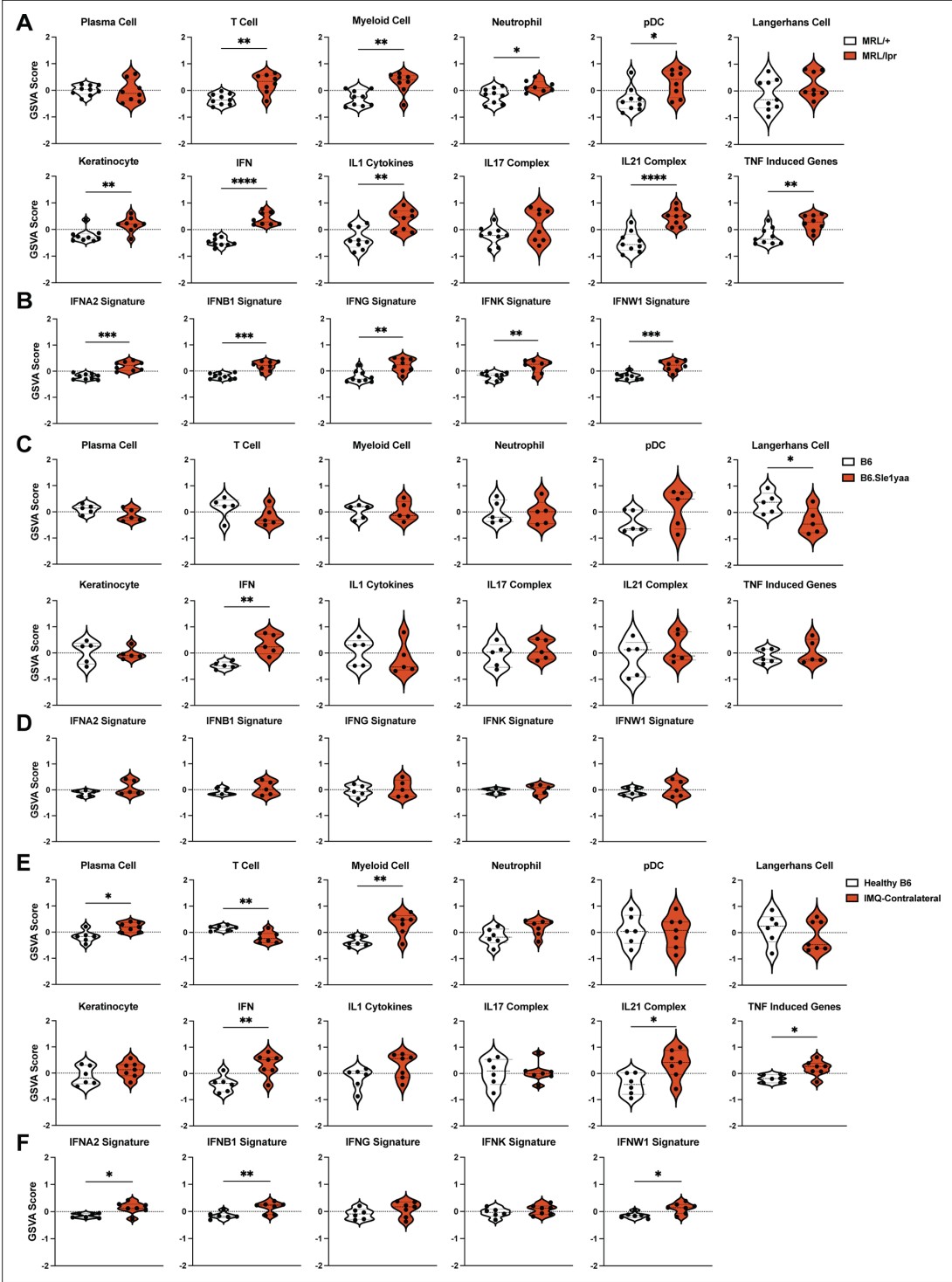

**Figure 3.** The SLE models share upregulated IFN signatures in non-lesional skin by gene set variation analysis (GSVA). (**A–F**) The RNAseq data from *Figure 2* of MRL/lpr (**A–B**), B6.Sle1yaa (**C–D**), and IMQ (**E–F**) models were analyzed by GSVA. (**A, C, E**) GSVA of gene sets relevant to lupus, adapted for murine models (**Kingsmore et al., 2021**). (**B, D, F**) GSVA of gene signatures specific to distinct IFN subtypes. (**A–F**) Each symbol represents one mouse. *p<0.05, **p<0.01, ***p<0.001, ****p<0.0001 by unpaired t-test.

The online version of this article includes the following figure supplement(s) for figure 3:

**Figure supplement 1.** GSVA of ipsilateral IMQ-painted skin together with contralateral ear skin.

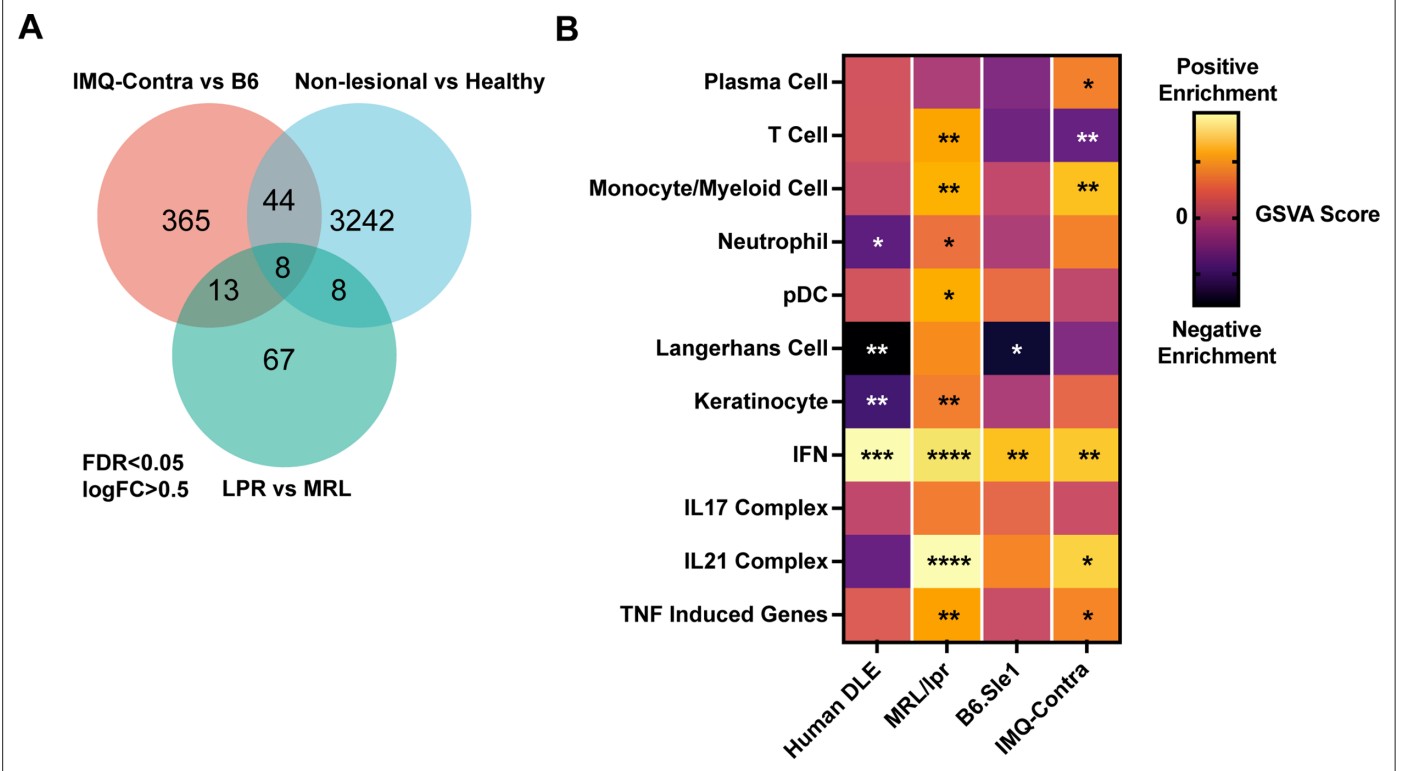

**Figure 4.** Comparison of human DLE and murine lupus models shows shared upregulation of IFN-associated genes. (**A**) Venn diagram of differentially expressed genes among DLE patients, LPR mice, and IMQ mice using FDR <0.05 and logFC >0.5. (**B**) Heatmap of GSVA scores for shared gene sets between non-lesional skin from DLE patients, LPR mice, B6.Sle1yaa mice, and IMQ mice. Asterisks indicate significant differences in GSVA scores compared to controls for each dataset.

(*Figure 3—figure supplement 1*). Overall, these data are consistent with the idea that, similar to human lupus skin, non-lesional skin in multiple SLE models is characterized by an IFN-rich environment that potentially primes the skin for photosensitive responses. These data also suggest that LCs sit within an IFN-rich environment in both human lupus and multiple murine models that may cause their dysfunction.

## IFN-I inhibits LC ADAM17 sheddase function

We examined the effects of IFN-I on UVR-induced LC ADAM17-mediated shedding and also sought to assess the extent to which altered sheddase activity reflected altered LC ADAM17 protein expression. We had previously established a facs-based LC ADAM17-mediated shedding assay by quantifying UVR-induced cell surface TNFR receptor 1 (TNFR1) loss at 45 min after UVR exposure, extensively validating that this loss reflected shedding activity, paralleled shedding of EGFR ligands into the supernatant, and was dependent on LC ADAM17 (*Shipman et al., 2018*). As we had studied LC ADAM17 sheddase activity using isolated LCs, we confirmed here that UVR treatment of a mixture of digested epidermal cells also induced LC cell surface TNFR1 loss (*Figure 5A*). This effect was LC ADAM17-dependent (*Figure 5A*), validating UVR-induced LC cell surface TNFR1 loss as a readout of LC ADAM17 activity in the context of a mixture of epidermal cells.

Anti-murine ADAM17 antibodies to quantify cell surface ADAM17 have not been available until one recently described by our group (*Lora et al., 2021*). This antibody was able to detect cell surface ADAM17 on LCs and other cells digested from skin, and this signal was specific, as it was lost from LCs in cells from mice lacking LC ADAM17 (*Figure 5B*; *Shipman et al., 2018*). Using the validated LC ADAM17 sheddase assay and new anti-murine ADAM17, we proceeded to examine the role of IFN-I in regulating LC ADAM17 activity and expression.

IFN-κ, an IFN-I that is highly expressed by lupus keratinocytes (*Tsoi et al., 2019*), reduced LC ADAM17 sheddase activity by nearly 40% when used to treat LCs ex vivo (*Figure 5C*). In contrast,

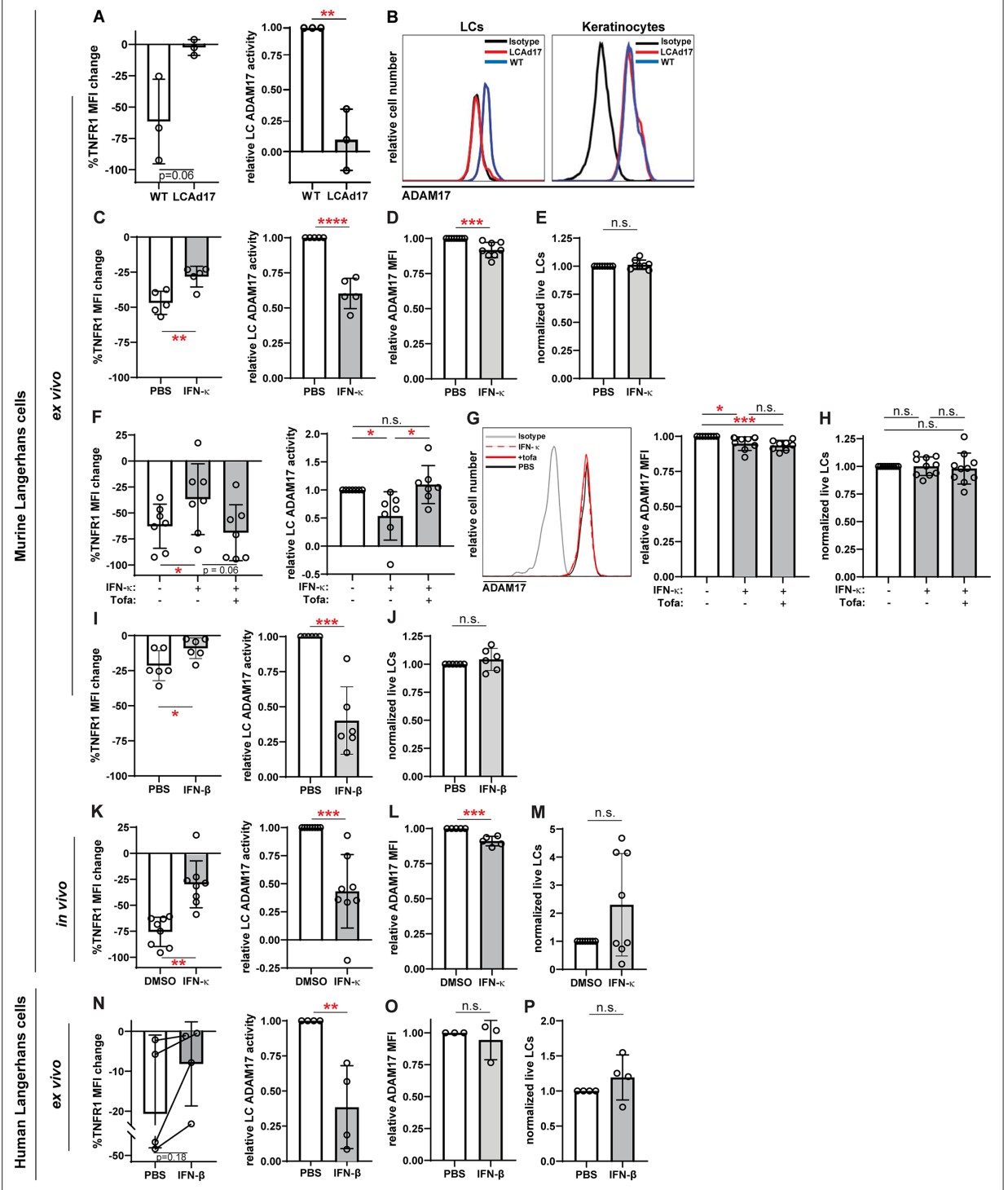

**Figure 5.** IFN-I inhibits LC ADAM17 sheddase activity. (**A**) Epidermal cell suspensions from WT and LCAd17 mice lacking ADAM17 in LCs were assayed for LC ADAM17 sheddase activity as indicated by the extent of UVR-induced cell surface TNFR1 loss. Percent change in cell surface TNFR1 mean fluorescence intensity (MFI) after UVR (left); relative LC ADAM17 activity calculated by normalizing TNFR1 loss to that of vehicle controls (right). (**B**) Representative histograms from cell surface ADAM17 staining of WT and LCAd17 epidermal cell suspensions. assayed. (**C–P**) Murine and human epidermal cell suspensions were treated ex vivo or mice were treated in vivo with IFN-I prior to assaying for UVR-induced LC ADAM17 sheddase activity (**C, F, I, K, N**), LC ADAM17 levels (**D, G, L, O**), and LC numbers (**E, H, J, M, P**). (**C–J**) Cells from WT mice were treated with IFN-κ or vehicle (**C–E**), with or without tofacitinib (**F–H**), or with IFN-β or vehicle (**I–J**). (**K–M**) IFN-κ or vehicle was applied topically to ears of WT mice 16–20 hr prior to examination. (**N–P**) Suction blister epidermal cell suspensions from healthy human donors were treated with IFN-β or vehicle. In (**N**), lines connect samples from the same donor. (**A,C–P**) Each symbol represents cells from a single mouse or donor, bars represent average values, and error bars are SD. n=3–10 per

*Figure 5 continued on next page*

Figure 5 continued

condition over 3–5 independent experiments. *p<0.05, **p<0.01, ***p<0.001, n.s.=not significant by paired (**A, C, F, I, K, N** (left)) or unpaired (**A, C, F, I, K, N** (right)), (**D–E, G–H, J, L–M, O–P**) t-test.

The online version of this article includes the following source data and figure supplement(s) for figure 5:

**Source data 1.** This file is the source file that contains all the data that led to the graphs in *Figure 5*.

**Figure supplement 1.** Langerhans cell yield and gating with suction blistering of human skin.

IFN-κ reduced LC ADAM17 protein levels only modestly by 8% and had no effect on cell viability (*Figure 5D–E*). The IFN-κ-driven inhibition of ADAM17 activity was abrogated by the JAK1/3 inhibitor tofacitinib (*Figure 5F*) without altering ADAM17 protein levels (*Figure 5G*) or cell viability (*Figure 5H*), supporting the idea that IFN-κ mainly affected LC ADAM17 by inhibiting sheddase activity. Another IFN-I, IFN-β, similarly reduced LC ADAM17 activity without effects on cell viability ex vivo (*Figure 5I–J*). In vivo, application of IFN-κ to the skin of wildtype B6 mice reduced LC ADAM17-mediated shedding by 57% (*Figure 5K*) while cell surface ADAM17 was reduced by 9% (*Figure 5L*). There was no effect on LC numbers (*Figure 5M*). The ex vivo and in vivo murine data together suggested that IFN-I downregulates UVR-induced LC ADAM17-mediated shedding mainly by inhibiting ADAM17 sheddase function rather than by reducing ADAM17 protein expression.

We also asked how IFN-I affects human LCs. While LCs can be obtained from skin biopsies or discarded tissues associated with surgical procedures, these tissues often require lengthy enzymatic digestion to dissociate the cells (*Shipman et al., 2018*), which has the potential to disrupt cell function. We found that epidermal roofs yielded from suction blistering could be digested for a short amount of time to obtain an epidermal cell mixture containing LCs (*Figure 5—figure supplement 1*; *Strassner et al., 2017*). Addition of IFN-β reduced LC ADAM17 activity by 62% without altering ADAM17 expression or LC viability (*Figure 5N–P*). Together, these results suggested that IFN-I can inhibit LC ADAM17 sheddase function in both murine and human LCs and supported the idea that high levels of IFN-I could potentially contribute to LC ADAM17 dysfunction in disease.

## IFNAR is important for LC ADAM17 dysfunction in multiple lupus models

We asked whether IFN-I was an important regulator of LC ADAM17 function in lupus models. Consistent with previous findings (*Shipman et al., 2018*), MRL/lpr mice showed a 42% reduction in LC ADAM17 sheddase activity when compared to control MRL/+ mice (*Figure 6A*). LC ADAM17 protein levels, however, were less affected, showing a 21% reduction (*Figure 6B*). Intraperitoneal (i.p.) anti-IFNAR treatment of MRL/lpr mice at 500 µg/dose restored UVR-induced LC TNFR1 loss (*Figure 6A*), suggesting restoration of LC ADAM17 sheddase activity. Anti-IFNAR did not alter cell surface ADAM17 protein expression (*Figure 6B*), suggesting that IFN-I regulated LC ADAM17 sheddase function rather than expression. A lower 125 µg/dose of anti-IFNAR had similar effects in restoring LC ADAM17 sheddase activity without affecting LC ADAM17 protein levels (*Figure 6C–D*). These results together suggested that IFN-I is important for inhibiting LC ADAM17 sheddase function in the MRL/lpr model.

LCs from the B6.Sle1yaa model showed a 63% reduction in UVR-induced TNFR1 loss when compared to control B6 mice without a corresponding change in cell surface ADAM17 protein levels (*Figure 6E–F*), suggesting reduced LC ADAM17 sheddase function in this SLE model. Similar to MRL/lpr mice, anti-IFNAR restored LC ADAM17 activity in B6.Sle1yaa mice without increasing ADAM17 protein expression (*Figure 6E–F*). These results suggest that IFN-I is an important inhibitor of LC ADAM17 sheddase function in the B6.Sle1yaa model.

Similar to the B6.Sle1yaa model, IMQ model mice showed reduced UVR-induced TNFR1 loss without alterations in cell surface ADAM17 protein levels when compared to controls (*Figure 6G–H*), suggesting reduced LC ADAM17 sheddase function. Similar to the MRL/lpr and B6.Sle1yaa models (*Shipman et al., 2018*) and consistent with the unchanged LC gene set by GSVA (*Figure 3E*), LC numbers were unchanged from controls in the IMQ model (*Figure 6I*). Anti-IFNAR restored LC ADAM17 sheddase activity in IMQ mice and did not alter LC numbers (*Figure 6G,I*). Together, these data suggesting that anti-IFNAR restored LC ADAM17 sheddase function in non-lesional skin across three distinct SLE models pointed toward a scenario whereby the IFN-I-rich environment in non-lesional lupus skin inhibits LC ADAM17 sheddase function.

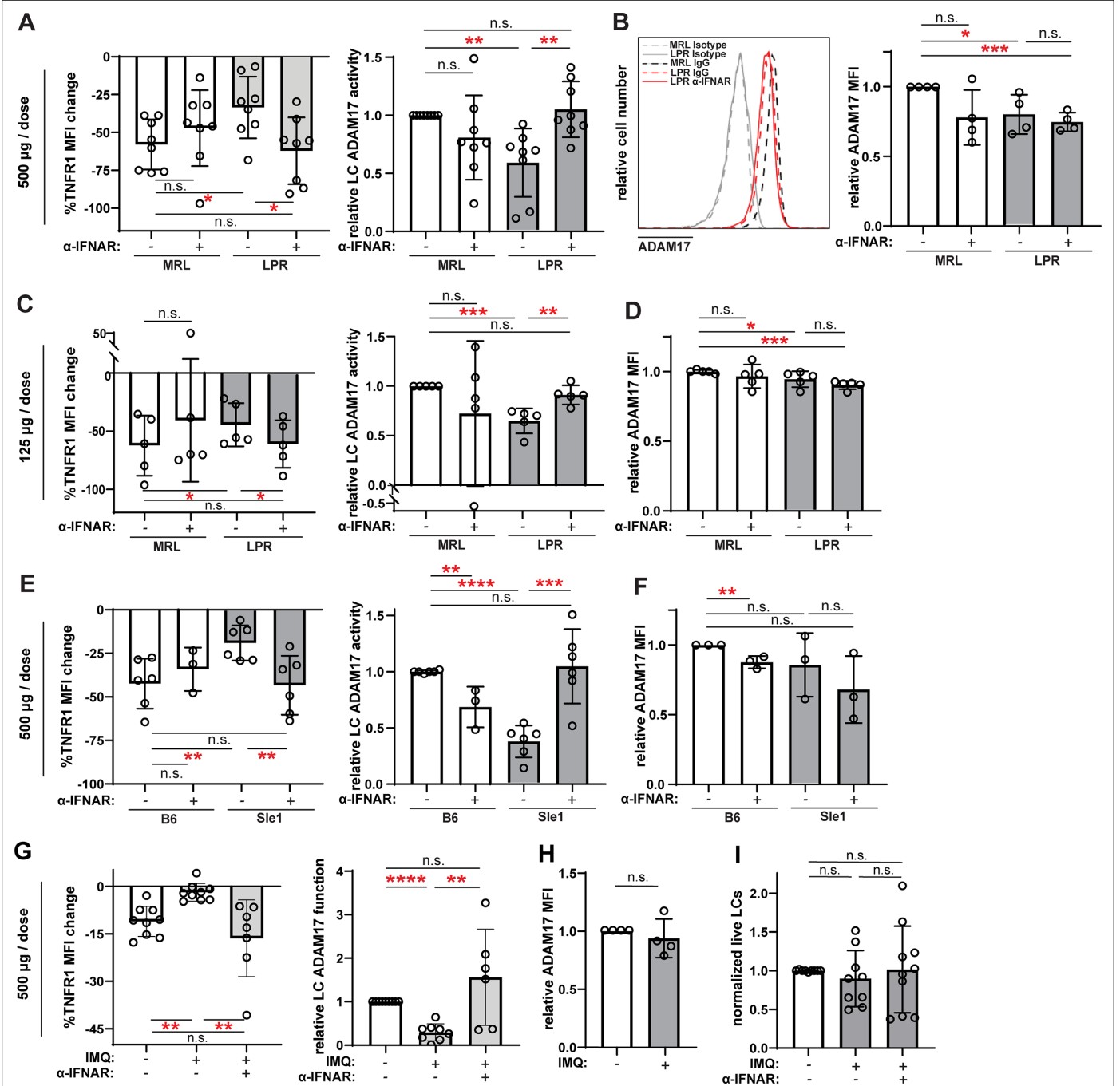

**Figure 6.** Anti-IFNAR restores LC ADAM17 activity in multiple lupus models. (**A–I**) MRL/lpr (**A–D**), B6.Sle1yaa (**E–F**), and IMQ (**G–I**) lupus model mice and their controls were treated twice with anti-IFNAR or isotype control at indicated doses over 6 days prior to collection of non-lesional epidermal cells. (**A, C, E, G**) UVR-induced LC ADAM17 sheddase activity as in **Figure 5**. (**B, D, F, H**) Relative cell surface ADAM17 levels. (**I**) Relative LC numbers. (**A–I**) Each symbol represents a mouse, bars represent average values, and error bars are SD. n=3–9 per condition over 3–8 independent experiments. *p<0.05, **p<0.01, ***p<0.001, n.s.=not significant by paired (A, C, E, G (left)) or unpaired (A, C, E, G (right)), (**B, D, F, H–I**) t-test.

The online version of this article includes the following source data for figure 6:

**Source data 1.** This file is the source file that contains all the data that led to the graphs in **Figure 6**.

**Table 1.** IFN inhibition of LC ADAM17 sheddase activity is uncoupled from LC numbers and LC ADAM17 surface expression.

Results from *Figures 5–6* and shows that IFN-mediated alterations in LC ADAM17 sheddase activity do not correlate consistently with changes in LC numbers or surface LC ADAM17 expression.

| | LC numbers | LC ADAM17 sheddase activity | LC ADAM17 cell surface levels |
|---|---|---|---|
| **In vitro IFN** | | | |
| Human skin +IFN | Unchanged | Down | Unchanged |
| Murine skin +IFN | Unchanged | Down | Modestly down |
| **In vivo IFN (murine)** | Unchanged | Down | Modestly down |
| **MRL/lpr model** | Unchanged | Down | Modestly down |
| Anti-IFNAR | | Restored | Unchanged |
| **B6.Sle.1yaa model** | Unchanged | Down | Unchanged |
| Anti-IFNAR | | Restored | Unchanged |
| **IMQ model** | Unchanged | Down | Unchanged |
| Anti-IFNAR | Unchanged | Restored | Unchanged |

We summarized the alterations in LC numbers, ADAM17 protein levels, and ADAM17 sheddase activity and effects of IFN-I modulation shown in *Figures 5–6* and *Shipman et al., 2018* to assess for consistent changes. While LC ADAM17 protein levels were not consistently affected and LC numbers were unchanged in the three lupus models examined, LC ADAM17 sheddase activity was consistently reduced (*Table 1*). Furthermore, manipulation of IFNAR signaling in these models, similar to treating human and murine LCs with IFN-I, consistently regulated only LC ADAM17 sheddase activity (*Table 1*). The uncoupling of IFN-I-mediated regulation of LC ADAM17 sheddase activity from LC numbers and LC ADAM17 expression suggests that IFN-I regulates LC ADAM17 sheddase function and points to an IFN-LC ADAM17 sheddase function axis as a potential mediator of IFN-I effects in skin.

## Anti-IFNAR reduces photosensitivity in a manner dependent on EGFR signaling and LC ADAM17

We next asked whether the anti-IFNAR-mediated restoration of LC ADAM17 sheddase activity impacted photosensitive skin responses. Anti-IFNAR treatment at 500 µg/dose reduced ear swelling in MRL/lpr mice (*Figure 7A–B*) but did not reduce epidermal permeability, a readout of skin barrier function (*Figure 7C*). Surprisingly, in control MRL/+mice, anti-IFNAR actually increased UVR-induced ear swelling and epidermal permeability (*Figure 7B–C*), suggesting that anti-IFNAR at this dose-induced photosensitivity in non-lupus mice. These results echoed findings by other groups that IFNAR deficiency in non-lupus B6 mice induced photosensitivity (*Sontheimer et al., 2017*) and reduced skin wound healing (*Gregorio et al., 2010*; *Wolf et al., 2022*), and we considered the possibility that the 500 µg anti-IFNAR dose perhaps mimicked complete IFNAR deficiency and disrupted normal skin functions, thus obscuring the effects of LC ADAM17 activity restoration. Reducing anti-IFNAR to 125 µg/dose still increased UVR-induced MRL/+ear swelling but no longer increased epidermal permeability (*Figure 7D–E*), suggesting a decreased pathologic effect on physiologic skin functions. This lower dose, which was sufficient to restore LC ADAM17 activity in MRL/lpr mice (*Figure 6C*), was sufficient to reduce both UVR-induced skin swelling and epidermal permeability (*Figure 7D–E*) and limit neutrophil and monocyte infiltration after UVR exposure (*Figure 7F*). Thus, anti-IFNAR at 125 µg/dose that restored LC ADAM17 sheddase function also reduced UVR-induced skin swelling, epidermal permeability, and inflammatory infiltrates in MRL/lpr mice.

We sought to understand the extent to which the anti-IFNAR-driven reduction in photosensitivity reflected the rescue of LC ADAM17 activity. As LC ADAM17 limits photosensitivity by stimulating EGFR (*Shipman et al., 2018*), we blocked EGFR signaling upon anti-IFNAR treatment of MRL/lpr mice (*Figure 7G*). The small molecule EGFR tyrosine kinase inhibitor PD168393 (PD16) abrogated the anti-IFNAR-mediated reduction in ear swelling and epidermal permeability (*Figure 7H–I*), suggesting that anti-IFNAR requires EGFR signaling to ameliorate photosensitivity in MRL/lpr mice. These data are

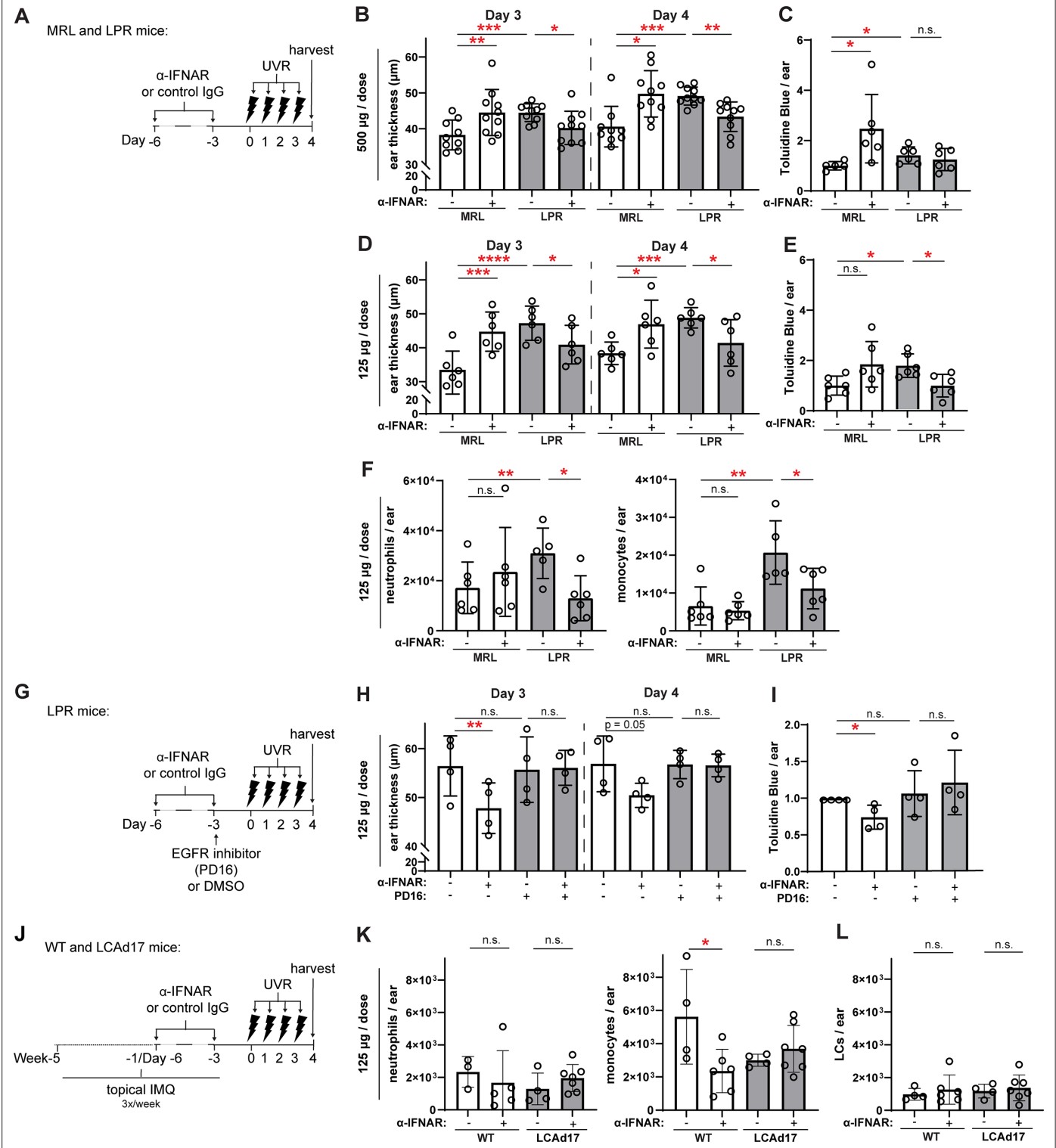

**Figure 7.** Anti-IFNAR reduces photosensitivity in EGFR- and LC ADAM17- dependent manners. (A–L) MRL/lpr (A–I) and LCAd17 (J–L) mice and controls were treated according to schematics in (A, G, J) and non-lesional ears were harvested. (B, D, H) Ear thickness. (C, E, I) Epidermal permeability as indicated by toluidine blue retention. (F, K) Neutrophils and monocytes, and (L) LCs per ear. (B–F, H–I, K–L) n=4–10 per condition over four to six independent experiments. *p<0.05, **p<0.01, ***p<0.001, n.s.=not significant by paired (B, D, H) and unpaired (C, E, F, I, K–L) t-test.

The online version of this article includes the following source data for figure 7:

**Source data 1.** This file is the source file that contains all the data that led to the graphs in *Figure 7*.

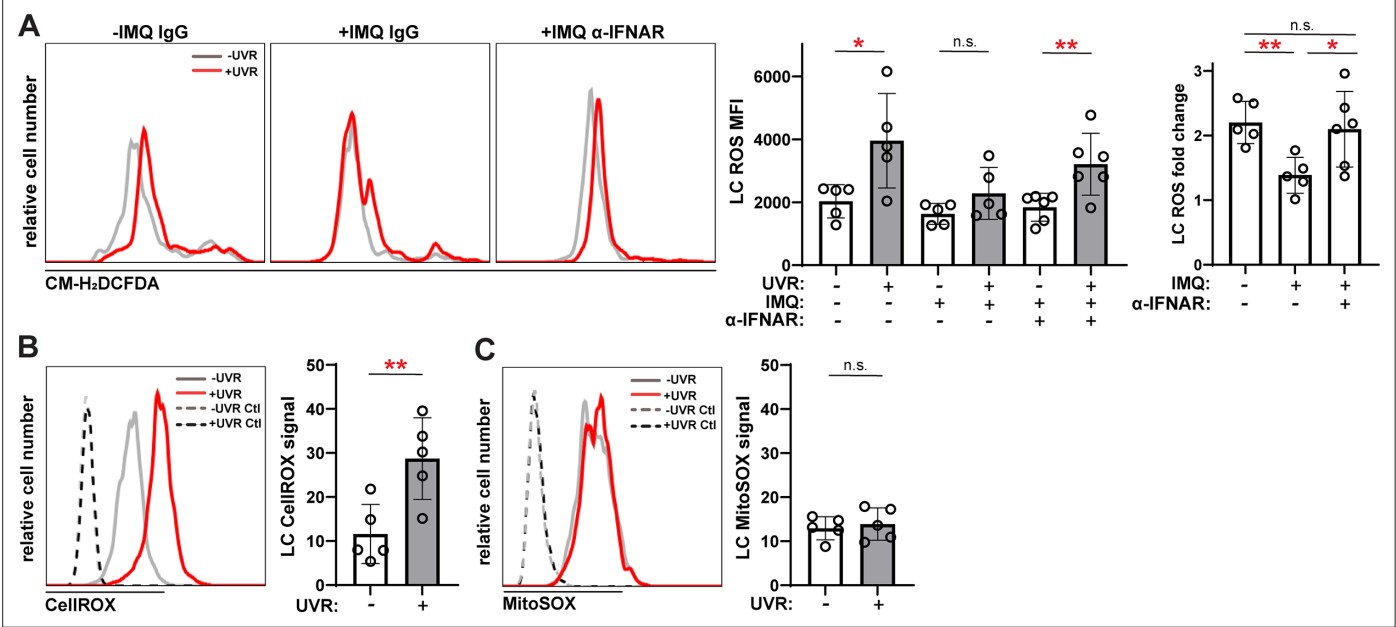

**Figure 8.** Anti-IFNAR restores UVR-induced LC ROS expression in a lupus model and UVR stimulates cytoplasmic ROS. (**A**) Epidermal cell suspensions from control or IMQ mice treated with IgG or anti-IFNAR were exposed to UVR. Cells were loaded with the general ROS indicator CM-H₂DCFDA prior to staining for LC markers for flow cytometry analysis. Representative histograms (left), CM-H₂DCFDA signal MFI (middle), and fold change with UVR exposure (right). (**B–C**) Healthy B6 epidermal cells were loaded with the cytoplasmic ROS indicator CellROX (**B**) or mitochondrial ROS indicator MitoSOX (**C**) prior to UVR exposure and staining of LC markers. Signal was calculated by dividing the ROS indicator MFI by the MFI of its respective negative control. Each symbol represents one mouse, bars represent average values, and error bars are SD. n=5–6 per condition over four to five experiments. *p<0.05, **p<0.01, n.s.=not significant by paired A (left), (**B, C**) and unpaired (A (right)) t-test.

The online version of this article includes the following source data for figure 8:

**Source data 1.** This file is the source file that contains all the data that led to the graphs in *Figure 8*.

consistent with the idea that anti-IFNAR reduces photosensitivity by restoring LC ADAM17 sheddase function.

IMQ mice were characterized by monocyte accumulation while neutrophils were lower in number (*Figure 7J–K*). Anti-IFNAR reduced the monocyte cell infiltrate in UVR-treated skin and had no effect on neutrophil numbers (*Figure 7K*). LC numbers were unaffected (*Figure 7L*). In IMQ-treated LCAd17 mice lacking ADAM17 in LCs (*Shipman et al., 2018*), however, anti-IFNAR failed to reduce monocyte accumulation (*Figure 7K*), suggesting that the ameliorative effects of anti-IFNAR on UVR-induced inflammation are, in part, dependent on LC ADAM17. That anti-IFNAR reduction of photosensitive responses across two lupus models depended on EGFR signaling and LC ADAM17 provided strong support for the idea that IFN-I contributes to photosensitivity at least in part by promoting LC ADAM17 sheddase dysfunction.

## IFN-I inhibits UVR-induced LC ROS expression

UVR has been shown to trigger ADAM17 sheddase activity in fibroblasts via ROS generation (*Singh et al., 2009*), and we asked if IFN-I modulation of ROS was associated with changes in LC ADAM17 activity. We observed the expected UVR-induced ROS generation in LCs from WT skin (*Figure 8A*; *de Jager et al., 2017*). Skin from IMQ-treated mice showed a reduction in UVR-induced LC ROS generation when compared to control skin, and anti-IFNAR treatment of IMQ mice restored UVR-driven ROS generation (*Figure 8A*). Assessment of cytoplasmic versus mitochondrial LC ROS showed that UVR exposure upregulated mainly cytoplasmic ROS (*Figure 8B–C*). These results suggested that IFN-I could potentially limit UVR-induced LC cytoplasmic ROS to inhibit LC ADAM17 sheddase function.

## Discussion

In this study, we demonstrate that non-lesional skin in three different murine SLE models is similar to human lupus skin in expressing an IFN signature and show that IFN-I contributes to the inhibition of LC ADAM17 sheddase dysfunction, which leads to photosensitivity. These results suggest that the IFN-rich environment in even non-lesional lupus skin promotes photosensitivity by at least in part by causing LC ADAM17 dysfunction and raises the possibility that the beneficial effects of anifrolumab on skin disease in SLE patients reflects the restoration of LC ADAM17 sheddase function. Our results also suggest that photosensitivity in the MRL/lpr, B6.Sle1yaa, and IMQ lupus models shares pathogenic mechanisms with human lupus, underscoring the utility of these models for the study of lupus skin pathophysiology.

Our transcriptomic data both expand on the characterization of non-lesional DLE skin and provides new characterization of non-lesional skin in MRL/lpr, B6.Sle1yaa, and IMQ lupus models. Comprehensive transcriptomic analyses of non-lesional lupus model skin has not been available previously and our data serve as a resource for understanding shared or distinct pathways between human lupus and murine models or among the distinct lupus models. Lupus patients are heterogeneous, which in part contributes to the difficulties in clinical trials (*Allen et al., 2021*). Understanding how specific lupus models can be used for studying different aspects of human lupus will help to advance disease insights and precision medicine approaches, especially as we consider how different tissues communicate with and regulate one another in disease.

Our data here, and especially the experiments showing the importance of EGFR signaling and LC ADAM17 in the ameliorative effects of anti-IFNAR on photosensitive responses, suggest that LC ADAM17 sheddase inhibition is a mechanism by which the IFN-rich environment primes non-lesional skin for photosensitive responses. This mechanism does not preclude parallel mechanisms such as potentiation of IL-6 expression and apoptosis of keratinocytes and activation of multiple other cell tyoes shown in the human system (*Sarkar et al., 2018*; *Stannard et al., 2017*; *Billi et al., 2022*). Given that LCs are positioned among the suprabasal layer of the epidermis near basal and spinous keratinocytes that demonstrate high IFN expression and signaling (*Billi et al., 2022*; *Psarras et al., 2020*; *Sarkar et al., 2018*), keratinocytes are likely to be an important source of IFN-I that causes LC ADAM17 sheddase dysfunction. This possibility coupled with our data here point to a potential pathogenic feed-forward loop between LCs and keratinocytes in lupus skin whereby keratinocyte-derived IFN-I causes LC ADAM17 sheddase dysfunction and this dysfunction contributes to greater UVR-induced keratinocyte damage, skin inflammation, and, over time, lesion formation.

Our findings suggest the possibility that IFNs have distinct functions in non-lesional and lesional skin. Physiologically, skin upregulates IFN-I with injury and infection, and the IFN is important in part for tissue repair and wound healing (*Gregorio et al., 2010*; *Wolf et al., 2022*; *Zhang et al., 2016*). This may explain why anti-IFNAR treatment of control MRL/+mice with physiologic IFN-I levels increased UVR-induced skin inflammation and suggests that the IFN-I in non-lesional skin in mic and humans reflects a repair response to a mild insult such as immune complex deposition that forms the 'lupus band' found in non-lesional human lupus skin. The upregulated IFN in non-lesional skin, by causing LC ADAM17 dysfunction or heightened keratinocyte responses, serves to 'prime' the skin for further injury and a second insult such as UVR manifests as a photosensitive response (*Billi et al., 2022*). Because the repair response is compromised, IFN signaling continues to be upregulated, further inhibiting repair responses and promoting inflammation, tissue damage processes, and lesion formation. In this model, then, IFN in non-lesional skin primes for further injury, and the (higher) IFN in lesional skin reflecting frustrated tissue repair, continues to prime but also contributes to tissue damage. IFNAR blockade, then, should both prevent new lesion formation and treat existing lesions as long as physiologic IFN levels for wound healing are preserved. Understanding the regulation and sources of the excess IFN-I in non-lesional and lesion skin will help to further develop therapies that ameliorate disease.

Our results extend the understanding of ADAM17 dysregulation as a contributor to disease. Increased ADAM17 activity can promote inflammation, such as by release of TNF or EGFR ligands within inflamed joints or lupus kidneys (*Issuree et al., 2013*; *Qing et al., 2018*); yet loss of ADAM17 can compromise barrier surfaces (*Blaydon et al., 2011*; *Franzke et al., 2012*). Here, we identify IFN-I as a negative regulator of LC ADAM17 sheddase function, and our data suggest a scenario whereby IFN-I reduces LC ROS generation to inhibit ADAM17 function. Further investigation will

better delineate the importance of LC ROS generation as well as contributions of other molecules such as the inactive Rhomboids (iRhoms) that regulate ADAM17 activity (*Geesala et al., 2019*).

This report importantly distinguishes among IFN-I regulation of LC ADAM17 sheddase function, LC ADAM17 surface molecule expression, and LC numbers. Our data here showing that the regulation of LC ADAM17 sheddase activity by IFN-I is uncoupled from LC ADAM17 protein expression and LC numbers (summarized in *Table 1*) supports the idea that LC ADAM17 sheddase function dysregulation is a key contributor to the propensity for photosensitivity and is thus a therapeutic target. The uncoupling of ADAM17 sheddase activity from protein expression is consistent with the known biology of ADAM17 on other cell types, where sheddase function is triggered by many stimuli on other cell types including UVR and G-protein-coupled receptors (GPCRs) and controlled by mediators such as ROS and iRhoms, as we and others have shown (*Li et al., 2015*; *Zunke and Rose-John, 2017*). We speculate that the ability to modulate ADAM17 enzymatic function offers the ability to regulate UVR responses in a more timely manner and with a much greater and tunable dynamic range than control of *Adam17* transcription and translation or LC numbers. Targeting the effects of IFN-I on LC ADAM17 sheddase function, then, offers the opportunity to have a large impact in protecting the skin and systemic complications.

A future direction will be to further understand LC functional phenotypes in human lupus and murine models. Our data along with recent reports indicates that LC loss is a feature of human lupus skin (*Billi et al., 2022*), with the effect being variable in non-lesional skin (*Martínez et al., 2022*; *Shipman et al., 2018*; *Sontheimer and Bergstresser, 1982*; this report) and a greater more consistent effect in lesional skin (*Bos et al., 1986*; *Martínez et al., 2022*; *Sontheimer and Bergstresser, 1982*). On the other hand, LC numbers appeared less affected in lupus mouse models, with reduction of the LC signature in the 5-month-old B6.Sle1yaa non-lesional skin and in the IMQ painted ear of IMQ mice, but otherwise with normal LC numbers in non-lesional skin of MRL/lpr, 10-month-old B6.Sle1yaa (*Shipman et al., 2018*) and IMQ mice (this report). Notably, the LC numbers and LC signature in non-lesional IMQ skin, despite being statistically unchanged from that of controls, is variable, suggesting LC loss in some mice and bearing similarities to the variability in human non-lesional skin. These data suggest similar trends between human and mouse models in LC loss, with mouse models perhaps being less severely affected. The extent to which LC ADAM17 sheddase dysfunction represents a milder or just a different dysfunction than loss of LCs, the regulation of LC numbers via regulation of survival, migration, replenishment, the consequences to skin and immune function, and similarities and differences between LCs in human disease and murine systems awaits further study.

High IFN-I states are also seen in viral infections, genetic interferonopathies, and other autoimmune diseases including rheumatoid arthritis and dermatomyositis (*Muskardin and Niewold, 2018*) in which photosensitivity is found in the former and is a disease hallmark in the latter (*Werth et al., 2004*; *Wysenbeek et al., 1989*). Autoimmune disease-like manifestations that include photosensitivity are among the immune-related adverse events in cancer patients treated with checkpoint inhibitors (*Brahmer et al., 2018*), a situation where distinct IFN-I-mediated processes may have specific roles in promoting anti-tumor immunity and treatment resistance (*Benci et al., 2019*). Our findings and the work that will follow will benefit patients with lupus and a wide spectrum of diseases.

# Materials and methods
## Study design

The purpose of the study was to understand how LC function is regulated by the environment in non-lesional skin to contribute to photosensitivity. The subjects are both humans and mice. We used gene expression analyses to understand the gene expression profiles in human lupus and murine lupus models. We used ex vivo flow cytometry-based assays, assessment of epidermal permeability, and skin swelling assays to understand the effects of IFN on LC function and implications for photosensitivity. Sample sizes and replicate numbers are provided in each figure legend. Sample sizes were determined based on previous similar types of experiments. Mice were randomly assigned to treatment groups. No criteria were set for excluding certain data points and no data points were excluded. No specific confounder variables such as order of treatments or cage location were controlled for except that our experiments used multiple independent cohorts of mice over replicate experiments.

Assessments and analyses were not blinded. ARRIVE reporting guidelines *Percie du Sert et al., 2020* have been used.

## Mice

Eight- to 16-week-old mice were used unless otherwise specified. Mice were sex and age-matched. For C57Bl/6 mice, LCAd17 (Langerin-cre; Adam17 f/f) mice (*Shipman et al., 2018*), and wildtype littermate controls, males and females were used. Unless otherwise indicated, for MRL/+and MRL/lpr mice, females were used at 8–12 weeks, while males were used at 12–16 weeks as females present with a disease phenotype earlier than males. For B6.Sle1yaa mice and their B6 controls, only males were used because the model depends on the autoimmune accelerator locus located on the Y chromosome. B6 mice were either bred in-house or purchased from Jackson Laboratory (Bar Harbor, ME). MRL/+, MRL/lpr mice, B6.Sle1yaa mice, and their B6 controls were either bred in-house or purchased from Jackson Laboratory. LCAd17 mice were bred in-house. Mice were kept in a specific pathogen-free barrier facility and all animal procedures including protocols to reduce pain, suffering, and distress were performed in accordance with the regulations of the Institutional Animal Care and Use Committee at Weill Cornell Medicine (New York, NY; protocol number 2015–0067).

## Mouse treatments

For IFN-κ painting, 37.5 ng IFN-κ (R&D Systems, Minneapolis, MN) was dissolved in DMSO (Sigma-Aldrich, St. Louis, MO) was applied topically onto each ear (18.75 ng on dorsal and ventral sides, each). For anti-IFNAR1 treatments, mice were injected IP with 500 µg/dose or 125 µg/dose of anti-IFNAR1 (MAR1-5A3) or isotype control (MOPC-21; Bio X Cell, West Lebanon, NH). For UVR treatments, mice were exposed to 1000 J UVB/m$^2$ (100 mJ/cm$^2$) daily for 4 consecutive days using a bank of four FS40T12 sunlamps that emitted a combination of UVA and UVB radiation, similar to described (*Shipman et al., 2018*). Timing of UVR exposure was made based on lamp emission of 1.31 mW/cm$^2$ UVB and distance of 9 cm away from lamp. To measure ear swelling after UVR exposure, a caliper (Mitutoyo, Aurora, IL) was used to take three measurements throughout each ear, and the reported value is the average of both ears. Each measurement was taken 22–24 hr after the previous measurement. For the IMQ-induced lupus mouse model, mice were painted on the dorsal and ventral sides of the right (ipsilateral) ear with 5% imiquimod cream (42 mg/ear/mouse) 3 x/week for 4–6 weeks (*Yokogawa et al., 2014*) prior to, and continued through experiments. The unpainted (contralateral) left ear was taken as non-lesional skin for flow cytometry experiments. For RNAseq, the painted ipsilateral ear and the unpainted contralateral 'non-lesional' left ear were taken and analyzed separately. Indicated MRL/lpr mice received 2.95 mg of the irreversible EGFR inhibitor PD168393 (0.74 mg on dorsal and ventral sides, each; Selleck Chem, Pittsburgh, PA).

## Human subjects

For microarray analysis, seven discoid CLE patients were examined. Both lesional and non-lesional skin were collected and the gene expression data from lesional tissue of these patients have been published, and patient characteristics are described (*Jabbari et al., 2014*). Healthy controls include the three that were previously described (*Jabbari et al., 2014*). All human tissue collection and research use adhered to protocols approved by the Institutional Review and Privacy Boards at the Rockefeller University (IRB# AJA-00740) and New York University (IRB# 10–02117), where participants signed written informed consents.

For suction blistering, one male and three female healthy participants between the ages of 23 and 55 participated. All human tissue collection and research use adhered to protocols approved by the Institutional Review and Privacy Board at the Hospital for Special Surgery (IRB# 2019–1998), where participants signed written informed consents.

## Human skin cell collection and preparation

Suction blistering was performed as previously described (27). Eight 5 mm suction blisters were generated with a negative pressure instrument (NP-4 model, Electronic Diversities, Finksburg, MD) on the arm of healthy donors over 30–60 min. The blister fluid was then collected by aspiration, and epidermal roofs were collected. After collection, epidermal roofs from suction blisters were digested with dispase (2.42 U/mL; Thermo Fisher Scientific, Waltham, MA), collagenase type II (616 U/mL;

Worthington Biochemical Corporation, Lakewood, NJ), and DNAseI (80 µg/mL; Sigma Aldrich, St. Louis, MO) for 40 min to generate single cell suspensions that were used for the ADAM17 sheddase assay.

## Skin cell collection and flow cytometry staining and quantification

For murine epidermal single cell suspensions, skin on the trunk was excised and subcutaneous fat scraped off. Skin was incubated in dispase (2.42 U/mL) at 37 °C. The epidermis was then gently peeled, finely minced, digested in type II collagenase (616 U/mL) and DNAse I (80 µg/mL).

For collection of ear skin, ears were cut along the cartilage ridge, and the dorsal and ventral sides were manually peeled. Ear skin was then finely minced and digested with a buffer containing dispase, type II collagenase, and DNAse I as described above. For murine LC TNFR1 staining for the ADAM17 sheddase activity assay (see below), epidermal cells were preincubated with Fc block and then stained so that we could gate on CD45+, CD3-, CD11c+, I-Ab + for B6 mice and I-Ak + for MRL mice, and EpCAM + LCs and assess for TNFR1. LCs were sometimes stained with rabbit-anti-ADAM17 (*Lora et al., 2021*) followed by donkey anti-rabbit IgG (Jackson Immunoresearch, West Grove, PA). Monocytes and T cells were identified by gating for CD45+, CD11b+, Ly6C+, and CD45+, CD11c-, I-Ak/I-Ab-, and CD3 + cells, respectively. 4',6-diamidino-2-phenylindole dihydrochloride (DAPI) was used to exclude dead cells and debris (Sigma Aldrich, St. Louis, MO). All antibodies are from Biolegend, San Diego, CA unless otherwise indicated.

For human LC TNFR1 staining for the ADAM17 sheddase activity assay, single cell suspensions of the epidermal roofs of suction blisters and blister fluid were preincubated with Fc block and subsequently stained for CD45+, CD11c+, CD14-, CD3- CD19-, CD56-, CD66b-, EpCAM+, and HLA-DR +to identify LCs. DAPI was used to exclude dead cells and debris, and TNFR1 was also stained to assess ADAM17 activity. All antibodies are from Biolegend, San Diego, CA unless otherwise indicated. Cells were analyzed using a FACS Canto or FACS Symphony A3 (BD Biosciences, San Jose, CA) and data were analyzed with FlowJo V10 Software (TreeStar, Ashland, OR).

For antibodies used, please see *Supplementary file 11*.

## Ex vivo ADAM17 sheddase activity assay

For murine skin, cell suspensions of epidermis were suspended at 200,000–500,000 cells/mL medium. For MRL/LPR and B6.Sle1yaa skin, medium was RPMI with 2% FBS, HEPES buffer, L-glutamine, penicillin-streptomycin (all Thermo Fisher Scientific, Waltham, MA). For IMQ mice skin, medium was phenol red-free RPMI (Corning, Edison, New Jersey), no serum, with HEPES buffer, L-glutamine, penicillin-streptomycin. Cells were exposed to 2000 (LPR and B6.Sle1yaa model mice) or 1000 (IMQ model mice) J/m$^2$ UVR incubated for 45 min, and then stained for cell surface markers described above. For IFN-κ treatment, 3.125 ng/mL murine IFN-κ (R&D Systems, Minneapolis, MN) was added to the epidermal cell suspension that were incubated for 16–20 hr prior to UVR exposure. Indicated cells were treated with 1 nM tofacitinib (Selleck Chem, Pittsburgh, PA) during IFN-κ incubation. For IFN-β treatment, 30 ng/ml murine IFN-β (R&D Systems) was added to the epidermal cell suspensions and assessed as above.

For human skin, cells were incubated for 1 hr with recombinant human IFN-β (35.7 ng/mL; activity = 2.8 × 10$^8$ UI/mg) and then exposed to 1000 J/m$^2$ UVR as above. Cells were further incubated for 45 min and collected for preincubation with Fc block. Cells were then stained for cell surface markers described above.

Calculating the relative change in cell surface TNFR1 levels upon UVR exposure was as previous (*Shipman et al., 2018*). The geometric MFI of TNFR1 was divided by the MFI for the isotype control to generate a relative MFI. The percent TNFR1 MFI change was calculated by setting the relative MFI of the control sample to 1 and normalizing the value from the experimental samples to that control. In experiments with more than one control sample, the control values were averaged, and the experimental samples were each normalized to the averaged value.

## Epidermal permeability assay

Epidermal barrier function was assessed as described (*Shipman et al., 2018*). Ear skin was dehydrated and rehydrated in graded methanol and then incubated in 0.1% toluidine blue dye (Sigma Aldrich, St. Louis, MO). The dye was extracted with a buffer consisting of 95% methanol, 2.5% sulfuric acid,

and 2.5% water and quantified with a colorimeter at 620 nm. Dye concentration was calculated using a standard curve.

## Ex vivo ROS assays

Epidermal cell suspensions in serum-free, phenol red-free DMEM/F-12 media were exposed to 1000 J/m2 UVR and the general ROS indicator 5-(and-6)-chloromethyl-2',7'-dichlorodihydrofluorescein diacetate, acetyl ester (CM-H$_2$DCFDA) (Invitrogen, Waltham, MA) at 10 uM was added immediately after UVR exposure. CM-H$_2$DCFDA oxidation by ROS inside the cells leads to fluorescence. Cells were further incubated for 45 min prior to cell surface marker staining to identify LCs by flow cytometry.

To assess cytoplasmic vs mitochondrial sources of ROS, epidermal cell suspensions were incubated with MitoSOX Green superoxide indicator (Thermo Fisher) and CellROX Deep Red Reagent (Thermo Fisher; final concentration of 5 µM for each) or vehicle (DMF and DMSO, respectively). Fifteen minutes after addition of reagents, cells were exposed to 750 J/m2 UVR, incubated another 15 min, and then washed three times with warm PBS prior to staining for extracellular surface markers for flow cytometry.

## RNA extraction

Female MRL/+and MRL/lpr 12–14 weeks old, 12-week-old B6 and B6.Sle1yaa male mice, and 12- to 14-week-old IMQ mice were used for RNA sequencing. RNA was extracted from ear skin using an RNEasy Kit (Qiagen, Germantown, MD) and quality confirmed on a BioAnalyzer 2100 (Agilent Technologies, Santa Clara, CA).

## RNA sequencing

Libraries were prepared with non-stranded and poly-A selection TruSeq RNA Sample Preparation kits. Sequencing libraries were sequenced by the Genomics Core Facility at Weill Cornell Medicine using a HiSeq 2500 (MRL/lpr, MRL/+samples) and NovaSeq 6000 (samples from B6.Sle1yaa and IMQ models and their controls) at a depth of 15–30 million mappable paired-end reads per sample.

## Computational analyses of gene expression data

Microarray analysis was performed using healthy controls and DLE lesional skin samples from an existing GEO dataset (GSE52471) obtained from GEO using GEOquery and combined with previously unpublished, unsubmitted non-lesional skin sample data obtained as part of the original study (*Jabbari et al., 2014*). Microarray data were normalized with gcRMA normalization using the gcrma R package (*Wu et al., 2021*). Differential gene expression analysis was performed with limma using the empirical bayes method (*Ritchie et al., 2015*) and controlling for microarray hybridization kit and sex. Batch correction was performed using ComBat (*Kammers et al., 2021*). Visualizations were generated using ComBat corrected data with plotly and ggplot2 in R.

For RNA sequencing analysis, read quality filtering and adapter trimming were performed using *fastp* (*Chen et al., 2018*). Filtered reads were mapped to the mouse genome (mm10), and exonic reads were counted against GENCODE release 27 annotation with the STAR aligner (*Dobin et al., 2013*) using a customized pipeline available at https://gitlab.com/hssgenomics/pipelines (*Oliver, 2023*). Differential gene expression analysis was performed with edgeR 3.30.2 and 3.38.4 using quasi-likelihood framework. Genes with low expression levels (<2 counts per million in at least one group) were filtered from all downstream analyses. p-Values were adjusted to correct for multiple testing using the Benhamini-Hochberg FDR procedure. Genes with adjusted p-values <0.05 were considered differentially expressed. Downstream analyses were performed in R and visualized using a Shiny-driven visualization platform (RNAseq DRaMA) developed at the HSS David Z. Rosensweig Genomics Research Center.

Differentially regulated pathways were identified using R QuSAGE 2.22.0 package (*Yaari et al., 2013*) with MSigDB C2 set (curated gene sets). All gene sets with less than 10 genes were excluded from the analysis. Pathways with Benhamini-Hochberg adjusted p-values <0.005 (non-lesional skin versus healthy control), <0.05 (MRL/lpr versus MRL/+, IMQ contralateral versus healthy B6) or unadjusted p-values <0.05 (Sle.yaa vs B6) were used for further analyses.

The R/Bioconductor package, GSVA (*Hänzelmann et al., 2013*) (v1.36.3), was utilized as a nonparametric, unsupervised method to estimate enrichment of pre-defined gene sets in gene expression

data from human DLE skin and non-lesional skin from MRL/lpr, B6.Sle1yaa, and IMQ mice. The inputs for GSVA were a matrix of expression values for all samples and curated gene sets describing select immune/tissue cell types and inflammatory cytokines. For the analysis of human DLE microarray data, low-intensity probes were filtered out if the interquartile range (IQR) of their expression values across all samples was not greater than 0 prior to the analysis. Enrichment scores were calculated using a Kolgomorov Smirnoff (KS)-like random walk statistic and represented the greatest deviation from zero for a particular sample and gene set. Scores across all samples were normalized to values between –1 (indicating no enrichment) and +1 (indicating enrichment). Gene sets used as input for GSVA are listed in *Supplementary files 3 and 9*. The human gene sets were previously published (*Martínez et al., 2022*). The mouse plasma cell, T cell, myeloid cell, neutrophil, pDC, dendritic cell, and keratinocyte gene sets were derived from Mouse CellScan, a tool for the identification of cellular origin of mouse gene expression datasets. The mouse IFN gene sets have been previously described (*Kingsmore et al., 2021*). The mouse cytokine signatures were generated by an iterative process involving derivation through literature mining and GO term definitions provided by the Mouse Genome Informatics (MGI) GO Browser (*Bult et al., 2019*).

## Statistical analyses

We determined the normality of data distribution using the Shapiro-Wilk test. For normally distributed data, we used two-tailed unpaired and paired Student's t-tests for comparisons between two conditions to evaluate p-values as indicated. For data that were not normally distributed, the Mann-Whitney test was used for unpaired and Wilcoxon matched-pairs signed rank test for paired comparisons. Significance was defined as p<0.05. GraphPad Prism software was used. For figures showing normalized values, the control sample was set to one, and the experimental samples were normalized relative to the control for that experiment. For experiments that contained more than one control sample, the mean was obtained for the control samples, and the individual control and experimental samples were calculated relative to this mean.

## Acknowledgements

The authors thank the members of the Lu Lab for helpful discussions. Alpha Omega Alpha Carolyn L Kuckein Student Research Fellowship (TML). Erwin Schrodinger Fellowship J 4638-B FWF (VZ). National Institutes of Health grant T32AR071302-01 to the Hospital for Special Surgery Research Institute Rheumatology Training Program (NS and WDS). National Institutes of Health grant MSTP T32GM007739 to the Weill Cornell/Rockefeller/Sloan-Kettering Tri-Institutional MD-PhD Program (WDS). Tow Foundation (YC and DJO). National Institutes of Health grant K08 AR069111 to the University of Iowa Department of Dermatology (AJ). Veterans Administration VA Merit I01 BX004907 (AJ), Physician Scientist Career Development Award from the Dermatology Foundation (AJ). National Institutes of Health grant R01AR077194 (AJ). National Institutes of Health grant R01 DK099087 (IR). National Institutes of Health grant R35GM134907 (CPB). National Institutes of Health grant R01AI079178 (TTL). National Institutes of Health grant R21 AR081493 (TTL). Department of Defense grant W81XWH-21-LRP-IPA (TTL). Lupus Research Alliance (TTL). St. Giles Foundation (TTL). Barbara Volcker Center for Women and Rheumatic Diseases (TTL). A Lasting Mark Foundation (TTL). National Institutes of Health Office of the Director grant S10OD019986 to Hospital for Special Surgery.

## Additional information

### Competing interests

Andrea R Daamen, Peter E Lipsky: is an employee of AMPEL BioSolutions, but has no financial conflicts of interest to report. Noa Schwartz: was awarded the Lupus Therapeutics: The Clinical Trial Network Infrastructure Grant, received by The Albert Einstein College of Medicine. The author received payment for lectures at the Congress of Clinical Rheumatology East and the Congress of Clinical Rheumatology West. The author has no other competing interests to declare. Jose Lora: has received the grants F31 NIH GM136144 and T32 NIH GM008539. The author has received stock or stock options from NASDAQ/NYSE Ticker: FULC, ABCL, AVXL, VOR, MRNA, BNTX, SAVA,

OCGN, CTMX, BCEL, GE. The author has no other competing interests to declare. William G Ambler: received support for travel and attending Lupus 21st century meeting in 2021. The author has no other competing interests to declare. Jonathan H Zippin: received a grant from NIH NIAMS, and consulting fees from Hoth Therapeutics and Pfizer. The author received payment for participation on a Data Safety Monitoring Board/ Advisory Board for Hoth Therapeutics and acts as President elect for PASPCR. The author holds stock options from Hoth Therapeutics, FoxWayne Inc and YouV labs. The author has no other competing interests to declare. James G Krueger: has received grant support from AbbVie, Akros, Allergan, Amgen, Avillion, Biogen, Botanix, Boehringer Ingelheim, Bristol-Myers Squibb, Exicure, Innovaderm, Incyte, Janssen, Kyowa Kirin, Lilly, Nimbus Lackshmi, Novan, Novartis, PAREXEL, Pfizer, Regeneron, UCB, Vitae Pharmaceuticals. The author received consulting fees from AbbVie, Aclaris, Allergan, Almirall, Amgen, Artax Biopharma, Arena, Aristea, Asana, Aurigene, Biogen Idec, Boehringer Ingelheim, Bristol-Myers Squibb, Escalier, Galapagos, Janssen, Kyowa Kirin, Lilly, MoonLake Immunotherapeutics, Nimbus, Novartis, Pfizer, Sanofi, Sienna Biopharmaceuticals, Sun Pharma, Target-Derm, UCB, Valeant, Ventyx. The author has no other competing interests to declare. Niroshana Anandasabapathy: has received the following grants: NIAMS AR080436-01, NIAMS R56AR078686-01 and NIH NIAMS 5R01 GRANT AR070234-05. The author received consulting fees from Immunitas, Shennon Bio and Janssen. The author received payment as a lecturer from 23 and me, Cellino and Bristol Meyer Squibb Genomics. They are also a board member of the Society of Investigative Dermatology. The author has no other competing interests to declare. Ali Jabbari: has received grants from the NIH and the VA and consulting fees from Pfizer. The author has no other competing interests to declare. Carl P Blobel: The patent number is US10024844B2 and the title of the patent is "Identification of an inhibitor of iRhom1 or an inhibitor of iRhom2", which is also what the patent relates to. Carl Blobel and the Hospital for Special Surgery have identified iRhom2 inhibitors and have co-founded the start-up company SciRhom in Munich to commercialize these inhibitors. Theresa T Lu: has received the following grants: NIH R01AI079178, NIH R21 AR081493, Department of Defense W81XWH-21-LRP-IPA, Lupus Research Alliance Lupus Innovation Award grant, Barbara Volcker Center for Women and Rheumatic Diseases grant. She has also received funding support from the St. Giles Foundation and A Lasting Mark Foundation. She has received consulting fees from Pfizer, and has a received payment from Bristol Meyers Squibb for giving a lecture. The author has received payment for attending Lupus 21st Century meeting. The author has no other competing interests to declare. The other authors declare that no competing interests exist.

## Funding

| Funder | Grant reference number | Author |
|---|---|---|
| Alpha Omega Alpha Honor Medical Society Carolyn L. Kuckein fellowship | | Thomas Morgan Li |
| HSS Medical Student Summer Research Fellowhship | | Thomas Morgan Li |
| Erwin Schrodinger Fellowship | J 4638-B FWF | Victoria Zyulina |
| NIH | T32AR071302 | Noa Schwartz William D Shipman |
| NIH MSTP grant | T32GM007739 | William D Shipman |
| Tow Foundation | | Yurii Chinenov David J Oliver |
| NIH | K08 AR069111 | Ali Jabbari |
| Veterans Administration VA Merit | I01 BX004907 | Ali Jabbari |
| Dermatology Foundation Physician Scientist Career Development Award | | Ali Jabbari |
| NIH | R01AR077194 | Ali Jabbari |

| Funder | Grant reference number | Author |
| --- | --- | --- |
| NIH | DK099087 | Inez Rogatsky |
| NIH | R35GM134907 | Carl P Blobel |
| NIH | R01AI079178 | Theresa T Lu |
| NIH | R21 AR081493 | Theresa T Lu |
| DOD | W81XWH-21-LRP-IPA | Theresa T Lu |
| Lupus Research Alliance | | Theresa T Lu |
| St. Giles Foundation | | Theresa T Lu |
| Barbara Volcker Center for Women and Rheumatic Diseases | | Theresa T Lu |
| A Lasting Mark Foundation | | Theresa T Lu |

The funders had no role in study design, data collection and interpretation, or the decision to submit the work for publication.

## Author contributions

Thomas Morgan Li, Conceptualization, Data curation, Formal analysis, Investigation, Writing – original draft, Writing – review and editing; Victoria Zyulina, Ethan S Seltzer, Marija Dacic, Formal analysis, Investigation, Writing – original draft, Writing – review and editing; Yurii Chinenov, Andrea R Daamen, Data curation, Formal analysis, Writing – original draft, Writing – review and editing; Keila R Veiga, Noa Schwartz, Ali Jabbari, Investigation, Writing – review and editing; David J Oliver, Data curation, Formal analysis, Writing – review and editing; Pamela Cabahug-Zuckerman, Investigation, Methodology; Jose Lora, Yong Liu, William D Shipman, William G Ambler, Jonathan H Zippin, Mehdi Rashighi, Niroshana Anandasabapathy, Methodology, Writing – review and editing; Sarah F Taber, Supervision, Writing – review and editing; Karen B Onel, Supervision; James G Krueger, Inez Rogatsky, Peter E Lipsky, Resources, Supervision, Funding acquisition, Project administration, Writing – review and editing; Carl P Blobel, Resources, Supervision, Funding acquisition, Methodology, Project administration, Writing – review and editing; Theresa T Lu, Conceptualization, Resources, Data curation, Formal analysis, Supervision, Funding acquisition, Writing – original draft, Project administration, Writing – review and editing

## Author ORCIDs

Thomas Morgan Li https://orcid.org/0000-0003-3503-3593
Ethan S Seltzer https://orcid.org/0000-0001-9675-9609
Noa Schwartz https://orcid.org/0000-0002-5577-3196
Yong Liu http://orcid.org/0000-0002-0196-3285
Jonathan H Zippin https://orcid.org/0000-0002-5882-0189
Inez Rogatsky http://orcid.org/0000-0003-3514-5077
Theresa T Lu https://orcid.org/0000-0002-5707-8744

## Ethics

For samples used in microarray analysis - human tissue collection and research use adhered to protocols approved by the Institutional Review and Privacy Boards at the Rockefeller University (IRB# AJA-00740) and New York University (IRB# 10-02117), where participants signed written informed consents. For samples used in suction blistering - human tissue collection and research use adhered to protocols approved by the Institutional Review and Privacy Board at the Hospital for Special Surgery (IRB# 2019-1998), where participants signed written informed consents.

All animal procedures were performed in accordance with the regulations of the Institutional Animal Care and Use Committee at Weill Cornell Medicine (New York, NY) (Protocol number 2015-0067).

## Decision letter and Author response

Decision letter https://doi.org/10.7554/eLife.85914.sa1
Author response https://doi.org/10.7554/eLife.85914.sa2

# Additional files

## Supplementary files

• Supplementary file 1. Differentially expressed genes in DLE vs healthy control non-lesional skin microarray.

• Supplementary file 2. Differentially expressed pathways in DLE vs healthy control non-lesional skin using QuSAGE.

• Supplementary file 3. Human gene sets used for GVSA.

• Supplementary file 4. Differentially expressed genes in MRL/lpr vs control MRL/+non-lesional skin RNAseq.

• Supplementary file 5. Differential expressed pathways in MRL/lpr vs control MRL/+non-lesional skin using QuSAGE.

• Supplementary file 6. Differentially expressed genes in B6.Sle1yaa vs control B6 non-lesional skin RNAseq.

• Supplementary file 7. Differentially expressed genes in IMQ vs control non-lesional skin RNAseq.

• Supplementary file 8. Differential expressed pathways in IMQ vs control non-lesional skin using QuSAGE.

• Supplementary file 9. Murine gene sets used for GVSA.

• Supplementary file 10. Common upregulated genes in human DLE, MRL/lpr mice, and IMQ mice.

• Supplementary file 11. Antibodies used.

• MDAR checklist

## Data availability

The murine RNAseq data have been deposited in GEO (GSE222573 for MRL/lpr and B6.Sle1yaa mice; GSE255519 for IMQ mice). Non-lesional human DLE microarray data were deposited with accession number GSE227329 with reanalysis of healthy control samples from GSE52471. All other data supporting the findings of this study are available within the paper and its Supplementary Materials.

The following datasets were generated:

| Author(s) | Year | Dataset title | Dataset URL | Database and Identifier |
|---|---|---|---|---|
| Schwartz N, Chinenov Y, Daamen AR, Oliver DJ, Lu TT, Li TM | 2023 | The interferon-rich skin environment regulates Langerhans cell ADAM17 to promote photosensitivity in lupus | https://www.ncbi. nlm.nih.gov/geo/ query/acc.cgi?acc= GSE222573 | NCBI Gene Expression Omnibus, GSE222573 |
| Schwartz N, Chinenov Y, Daamen AR, Oliver DJ, Dacic M, Jabbari A, Rogatsky I, Lu TT, Li TM | 2024 | The interferon-rich skin environment regulates Langerhans cell ADAM17 to promote photosensitivity in lupus | https://www.ncbi. nlm.nih.gov/geo/ query/acc.cgi?acc= GSE255519 | NCBI Gene Expression Omnibus, GSE255519 |
| Schwartz N, Jabbary A, Oliver D, Chinenov Y, Lu T, Li TM | 2023 | The interferon-rich skin environment regulates Langerhans cell ADAM17 to promote photosensitivity in lupus [array] | https://www.ncbi. nlm.nih.gov/geo/ query/acc.cgi?acc= GSE227329 | NCBI Gene Expression Omnibus, GSE227329 |

The following previously published dataset was used:

| Author(s) | Year | Dataset title | Dataset URL | Database and Identifier |
|---|---|---|---|---|
| Jabbari A, Suárez-Fariñas M, Krueger JG | 2013 | Dominant Th1 and Minimal Th17 Skewing in Discoid Lupus Revealed by Transcriptomic Comparison with Psoriasis | https://www.ncbi. nlm.nih.gov/geo/ query/acc.cgi?acc= GSE52471 | NCBI Gene Expression Omnibus, GSE52471 |

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
