## [Editor Report]

This study presents a useful assessment of the possible role of type I interferons in inhibiting Adam17 protease/sheddase activity and their correlation with decreased Langerhans Cells signature in lesional and nonlesional CLE and murine models as cause of photosensitive lupus. The data were collected and analyzed using solid methodology. This work will be of interest to scientists interested in photosensitivity in the setting of lupus.

---

## [Decision Letter]

**Decision letter after peer review:**

Thank you for submitting your article "The interferon-rich skin environment regulates Langerhans cell ADAM17 to promote photosensitivity in lupus" for consideration by *eLife*. Your article has been reviewed by 3 peer reviewers, one of whom is a member of our Board of Reviewing Editors, and the evaluation has been overseen by Betty Diamond as the Senior Editor.

Essential revisions (for the authors):

1) The authors present the data on TNFR1 MFI change in the three different lupus-prone mice but no differences were detected in the actual ADAM17 protein expression. Quantifying changes in Adam17 activity in the photosensitive models in relation to Adam17 levels and numbers of LCs, both of which are affected by UV exposure, would better establish the proposed axis. Knock out mice for ADAM17 or a blocking antibody for Adam17 with assessment of the Langerhans cell function would be helpful to directly prove the hypothesis.

2) The authors should try to link their data to the existing literature and validate their results by using human samples, as not all murine lupus models have a strong interferon-mediated disease, as with the MRL/lpr mouse model. Also, no difference in LC numbers was reported with the B6.Sle1yaa lupus-prone model.

3) The authors suggest that LC ROS is reduced in lupus model mice and restored by anti-IFNAR treatment. Measurement of an increase in oxidative phosphorylation (OXPHOS) in the cells triggered by UVR would be helpful. LCs from actual lupus patients are not used in the experiment.

*Reviewer #1 (Recommendations for the authors):*

It would be helpful to discuss why there is such a focus on nonlesional skin, when lesional biopsies show an even greater decrease in LCs. They also have higher levels of IFN that in nonlesional skin. Also, lesional skin is photosensitive.

Line 195-197. It is stated that there is a loss of epidermal LC and keratinocyte function in the context of an IFN-rich pro-inflammatory environment, leading to photosensitivity in nonlesional lupus skin. The data shown in Figure 1 shows that upregulation of IFNs is significantly higher in lesional than nonlesional skin. It would be helpful to discuss the implications of the lesional data also.

It would be helpful to compare lesional and nonlesional in terms of the signatures shown in Figure 1 G and H. Is nonlesional different in terms of the IFN subtypes relative to lesional skin?

Langerhans cells were lower in lesional than nonlesional DLE, so not sure the parallel with early diseased mice B6.Sle1yaa mice and DLE is correct. Presumedly established DLE is more like the older mice who didn't show a difference in Langerhans cells once the disease was established. Would suggest revising this section a bit.

*Reviewer #2 (Recommendations for the authors):*

Non-lesional skin in lupus model mice share an IFN-I signature and other functional gene expression modules with human lupus skin:

1. Lesional and non-lesional single-cell RNA seq analysis was recently published by Bili et al. (https://doi.org/10.1126/scitranslmed.abn2263).

2. Important literature is missing: Keratinocytes in non-lesional lupus skin produce IFN-kappa (https://doi.org/10.1038/s41584-022-00826-z). Upregulation of interferon-stimulated genes (ISGs) was already described in the blood and non-lesional skin of ANA-positive individuals. This ISG upregulation was indeed lower compared to lesional skin from SLE patients (https://doi.org/10.1038/s41467-020-19918-z).

3. The MRL/lpr lupus-prone model has a less type I IFN-mediated disease, so it is not surprising that the authors found less significant changes compared to the control model.

4. The results from the B6.Sle1yaa lupus-prone model are equally less convincing. No difference in LC numbers was reported, whilst the authors only found Cd207 expression downregulation in 5-month deceased mice. These results are hard to interpret and to be associated with pathology in humans.

5. The results of ISG upregulation in the imiquimod (IMQ) model have been described in the literature many times; it is well established that imiquimod can induce an interferon response via TLR7 activation and upregulation of IRF target genes such as Irf7, Irf8 and Stat1.

6. The authors report that Isg15 was upregulated in the non-lesional ears of the IMQ mice, but the interpretation of the results needs more careful consideration. Isg15 was shown to be upregulated and secreted by plasmablasts in lupus patients inducing antibody-independent inflammation (https://doi.org/10.4049/jimmunol.1600624).

7. The authors conclude that "These data are consistent with the idea that non-lesional skin is primed for photosensitive responses and that LCs sit within an IFN-rich environment in both human lupus and multiple murine models that may cause LC dysfunction". Non-lesional skin has been shown to contribute to IFN-mediated responses as cited above. However, the authors do not provide robust in vitro or in vivo confirmation about how this IFN response mediates distinct phenotypes of LCs.

IFN-I inhibits LC ADAM17 function

1. The authors assessed LC ADAM17 function by quantifying UVR-induced cell surface TNFR receptor 1 (TNFR1) shedding as "previously described". They need to provide insights about their methodological approach and not just simply cite a previously published article.

2. The authors present data in Figure 4 only for the WT mice and some human LCs. LCs do not seem to be isolated from skin biopsies of lupus patients, whilst IFN-kappa is used for murine LCs but IFN-β for human LCs. Why is there a different approach to choosing other type I IFNs?

IFNAR is important for LC ADAM17 dysfunction in multiple lupus models

1. The authors present the data on TNFR1 MFI change in the three different lupus-prone mice but no differences were detected in the actual ADAM17 protein expression. These results are difficult to interpret. Have the authors considered knocking out the mice for ADAM17 and reassessing the function of LCs?

Anti-IFNAR reduces photosensitivity in an EGFR and LC ADAM17-dependent manner

1. The authors suggest that anti-IFNAR increased photosensitivity in non-lupus mice. How is this related to human data clearly showing that UVR is linked to photosensitivity by inducing IFN-kappa expression by keratinocytes both in vitro and in vivo (http://dx.doi.org/10.1136/annrheumdis-2018-213197 and https://doi.org/10.1038/s41467-020-19918-z)?

IFN-I inhibits UVR-induced LC ROS expression

1. In figure 7, the authors suggest that LC ROS is reduced in lupus model mice and restored by anti-IFNAR treatment. They use ROS indicator but they have not tried to measure if there is an increase in oxidative phosphorylation (OXPHOS) in the cells triggered by UVR. LCs from actual lupus patients are not used in the experiment.

*Reviewer #3 (Recommendations for the authors):*

– The analysis of the publicly available human data is extensive and largely confirms the already published findings of high IFN signatures in non-lesional and lesional DLE skin. While it is encouraging that the overall analyses of IFN-I response and immune cell signatures are consistent with already published findings, the sheer amount of data distracts from the finding most relevant to this study: decreased Langerhans Cell signature in non-lesional and lesional skin. The reviewer suggests the data in Figure 1 be summarized in 1 or 2 more comprehensive panels (e.g. a heatmap of cell signatures or cytokine signatures).

– Similar to the human data, the findings of differential gene signatures in the skin of the murine lupus models could be summarized in a more comprehensive manner. The amount of data distracts from the most significant findings, particularly in B6.Sle1yaa model: decreased DC signature and increased IFN signature. Likewise, the reader is left to wonder at the end of Figure 2 if the IMQ-induced IFN response is accompanied by a decrease in LC numbers and/or function.

– Summarizing the data as the IFN-I signature may be useful in Figure 2J to provide a better insight into the IFN-I response overall.

– Computational analyses in Figures 1 and 2 emphasize the co-occurrence of a high IFN-I signature and a low LC and/or DC signature. It is not clear if the downregulation of the DC gene set indicates diminished presence of LCs in the non-lesional skin of the lupus mouse models or "reflects decreased LC function" as the authors suggest.

– The significance of the Th1 signature in the CLE cohort discussed in line #149 to the interferon signature is unclear.

– Unclear what figure the findings in lines #175-178 refer to?

– Unclear what figure the findings in lines #310-315 refer to?

– The specificity of ADAM17 for TNFR1 is not explicitly stated. The previously published data using the LCad17 mouse should be emphasized and some of the supplemental data included in the main figures.

– In Figure 7 it is not clear whether and how the ROS are being specifically measured in LCs.

– Given the hypothesis that IFN-I may be the cause of a decreased DC signature in the mouse skin, it would be relevant to ask if this signature is also decreased in the IMQ model, which is a known model of IFN-induction as confirmed by the authors. Likewise, asking how anti-IFNAR treatment affects the DC signature / LC numbers would be important, in the absence and presence of UV. The authors indicate in Figure 5I that IMQ reduces LC numbers.

– Decreased inflammation in LCad17 mice in the IMQ+UV model is unexpected. Previous studies by this group showed increased UV-induced inflammation in the absence of LC-ADAM17 (Shipman et al. 2018). Is it possible that IMQ treatment prior to UV sets off a negative feedback loop that counteracts absence of Adam17? If the authors treated LCad17 mice with anti-IFNAR IgG prior to UV in the absence of IMQ, would the results defer? UV light is an inducer of IFN itself so may serve as the relevant stimulus.

– UV light is an important inducer of IFN. Authors have previously shown that UV also induces Adam17 expression. Therefore, the question remains whether a high baseline IFN signature in lupus skin suppresses UV-induced Adam17 expression?

– A direct mechanistic link between high IFN-I and loss of Adam17 activity driving photosensitive reactions could be strengthened. Would blocking Adam17 with a blocking antibody suppress photosensitive reactions in lupus mouse models? Would treating LCAd17 mice with IFN fail to enhance or diminish UV-induced inflammation?

[Editors' note: further revisions were suggested prior to acceptance, as described below.]

Thank you for resubmitting your work entitled "The interferon-rich skin environment regulates Langerhans cell ADAM17 to promote photosensitivity in lupus" for further consideration by *eLife*. Your revised article has been evaluated by Betty Diamond (Senior Editor). We apologize that the original Reviewing Editor has become unavailable.

The manuscript has been improved but there are some remaining issues that need to be addressed, as outlined below. Please respond to this as quickly as possible; it should not be necessary to consult reviewers again for the final revision.

Reviewer 2 remarks that IFN enrichment seems too high for all clusters. Can you please more fully explain your conclusion regarding the IFN signature?

*Reviewer #2 (Recommendations for the authors):*

The authors have made a significant effort to address the comments pointed in the first review by reanalysing data, introducing new results, and updating essential literature missing.

However, I would like to express some concerns about to original validity of the data. The authors state in the revised manuscript that "We had mistakenly shown lesional DLE data rather than non-lesional data for Figures 1A-D in our original submission". It would be impossible for the reviewing process to validate all data presented and a good will is expected. The authors should be accountable for presenting the right data and results in the first place.

The authors also state that "they have replaced the limited qPCR data with RNAseq data (Figure 2F-I, Figure 3E-F) to gain a more complete picture of the IFN signature and gene expression in this model". From the data presenting in Figure 4, I can see how "the non-lesional skin shares key pathways with non-lesional skin of human DLE and other lupus models (Figure 4B), while the IMQ-painted skin surprisingly did not yield an IFN signature (Figure 3—figure supplement 1), potentially reflecting tissue damage or some other distinct biology that reflects the prolonged direct exposure to IMQ". IFN-enrichment appears to be high in all clusters.

I would strongly recommend that the revised manuscript requires an expert bioinformatic review to analyse the computational approaches and reported results.

---

## [Author Response]

Essential revisions (for the authors):1) The authors present the data on TNFR1 MFI change in the three different lupus-prone mice but no differences were detected in the actual ADAM17 protein expression. Quantifying changes in Adam17 activity in the photosensitive models in relation to Adam17 levels and numbers of LCs, both of which are affected by UV exposure, would better establish the proposed axis. Knock out mice for ADAM17 or a blocking antibody for Adam17 with assessment of the Langerhans cell function would be helpful to directly prove the hypothesis.

We have now (1) made a point of distinguishing between IFN regulation of LC ADAM17 sheddase activity vs LC ADAM17 cell surface expression and LC numbers in our Introduction and description of (Figures 5-6), and (2) added Table 1 summarizing the effect by IFN on LC ADAM17 sheddase activity vs LC ADAM17 cell surface expression and LC numbers described in Figures 5-6. The table shows that regulation by IFN-I of LC ADAM17 sheddase activity is uncoupled from LC ADAM17 protein expression and LC numbers, consistent with the idea that IFN (negatively) regulates LC ADAM17 sheddase function to contribute to photosensitivity. The uncoupling of ADAM17 sheddase activity from protein expression is consistent with the known biology of ADAM17, where sheddase function is triggered by a wide variety of stimuli on other cell types including ultraviolet radiation and GPCRs and controlled by mediators such as by ROS and iRHOMs, as established in part by the work of co-author Carl Blobel. We speculate that the ability to modulate ADAM17 enzymatic function offers the ability to regulate UVR responses in a more timely manner and with a much greater and tunable dynamic range than control of ADAM17 transcription and translation or LC numbers. Targeting the effects of IFN-I on LC ADAM17 sheddase function, then, offers the opportunity to have a large impact in protecting the skin and systemic complications. (Text lines 366-375, 487-501 as well as text scattered throughout (66-67, 69-72, 105-113, 323-325, 345, 354-355, 361-365)).

We also moved data from the Supplementary Materials to new Figure 5A validating the LC ADAM17 shedding assay. We had previously established a facs-based LC ADAM17-mediated shedding assay by quantifying UVR-induced cell surface TNFR receptor 1 (TNFR1) loss at 45 minutes after UVR exposure, extensively validating that this loss reflected shedding activity, paralleled shedding of EGFR ligands into the supernatant, and was dependent on LC ADAM17 (Shipman et al., SciTransMed 2018,10(454):eaap9527). As we had studied LC ADAM17 sheddase activity using isolated LCs, we confirm in Figure 5A that UVR treatment of a mixture of digested epidermal cells also induced LC cell surface TNFR1 loss. This effect was LC ADAM17-dependent, validating UVR-induced LC cell surface TNFR1 loss as a readout of LC ADAM17 activity in the context of a mixture of epidermal cells. (Text lines 298-306).

2) The authors should try to link their data to the existing literature and validate their results by using human samples, as not all murine lupus models have a strong interferon-mediated disease, as with the MRL/lpr mouse model. Also, no difference in LC numbers was reported with the B6.Sle1yaa lupus-prone model.

In this extensively revised manuscript, we have included several additional references and expanded our description of literature documenting an IFN-rich environment in non-lesional human skin in the Introduction (lines 118-128). The consistency of this finding across SLE, CLE, and pre-clinical autoimmunity, along with the beneficial effects of anifrolumab on lupus skin, speak to the contribution of IFN-I to lupus skin disease. We show using human cells that IFN-I reduces LC ADAM17 sheddase function without affecting LC ADAM17 protein levels or LC numbers (Figure 5N-P), supporting the idea that IFN inhibits LC ADAM17 activity to contribute to photosensitivity.

We have added new RNAseq data of IMQ skin (Figure 2F-I, Figure 3E-F, Figure 2—figure supplement 1B, Figure 3—figure supplement 1) to show that all three murine SLE models examined are remarkably consistent in demonstrating upregulated IFN signatures in non-lesional skin, strengthening the idea that the models share pathogenic mechanisms with human lupus. Note that Vital and colleagues have shown that the enrichment of an IFN signature in skin is far greater than the enrichment in blood in SLE and at-risk patients (Psarras et al. 2020, Nature Comm 11:6149), indicating that skin is a specific IFN-rich niche, and the clear IFN signature in LPR skin and functional data could reflect the uniqueness of the skin. This shared IFN signature in non-lesional skin in humans and mice, the evidence in humans that IFN is a driver of photosensitivity, and our data showing that IFN-I inhibits LC ADAM17 sheddase function in both human and murine LCs, that anti-IFNAR reduces photosensitive skin responses in the lupus models and does so in an EGFR- and LC ADAM17-dependent manner together strongly support the idea that IFN-I inhibits LC ADAM17 sheddase function to contribute to photosensitivity.

We also note that in-depth transcriptomic analysis of murine model non-lesional skin has not been previously done, and that the RNAseq data in the MRL/lpr, B6.Sle1yaa, and Imiquimod models show shared and unique features with human CLE data and each other. These data should serve as a good resource for investigators interested in using these models for studying different aspect of lupus skin disease.

The reduced LC gene set in B6.Sle1yaa non-lesional skin in the 5 month old mice examined in our current report (Figure 3C) and the unchanged LC numbers in the 10 month old mice we examined in (Shipman et al., SciTransMed 2018 ,10(454):eaap9527) may reflect a loss of LCs at 5 months of age but not at 10 months of age or it may reflect normal LC numbers but an altered LC phenotype with downregulated LC expression of CD207 and other genes in the LC gene set. With regard to the possibility of different LC numbers at 5 and 10 months of age, LC numbers are known to demonstrate a physiologic reduction with age, and a potential scenario is that LCs are reduced in 5 month old B6.Sle1yaa mice (compared to controls) but the physiologic decline in the WT controls by 10 months of age equalizes LC numbers in WT and B6.Sle1yaa mice. We do now comment on LC numbers in human and mice in our Discussion (lines 502-518), but our main focus in this extensively revised manuscript is, as described in Point #1, that our data clearly establish that IFN-I consistently regulates LC ADAM17 sheddase activity (rather than LC ADAM17 expression or LC numbers). These data point to an IFN-I-mediated regulation of LC ADAM17 sheddase function that contributes to IFN-I effects in skin.

3) The authors suggest that LC ROS is reduced in lupus model mice and restored by anti-IFNAR treatment. Measurement of an increase in oxidative phosphorylation (OXPHOS) in the cells triggered by UVR would be helpful. LCs from actual lupus patients are not used in the experiment.

We agree that our original intracellular ROS staining did not indicate whether the ROS is cytoplasmic or mitochondrial in origin and UVR triggering of increased OXPHOS would suggest the involvement of mitochondria ROS. We now add Figure 8B-C using mitochondrial and cytoplasmic ROS indicators. The data show that UVR stimulates an early increase in cytoplasmic but not mitochondrial ROS. This is consistent with the critical role of cytoplasmic catalase in early UVB-induced ROS generation in keratinocytes (Heck et al. JBC 2003. 278:22432).

Reviewer #1 (Recommendations for the authors):It would be helpful to discuss why there is such a focus on nonlesional skin, when lesional biopsies show an even greater decrease in LCs. They also have higher levels of IFN that in nonlesional skin. Also, lesional skin is photosensitive.

Thank you for your thoughtful review. We have focused on non-lesional skin because we are interested in understanding the factors that contribute to the propensity to photosensitivity and are thus interested in the mechanisms that have already gone awry in even non-lesional skin. While lesional skin also provides a lot of clues about pathophysiology, it is more difficult to dissociate the initial underlying dysfunction from the effects of tissue damage and contributing effector mechanisms. In this revision, we have expanded on discussion of the lesional data of Figure 1 (lines 165-177) and added to the Discussion on potential distinct roles for IFN in non-lesional and lesional skin (lines 460-477). We also now state in the Introduction why we focus on non-lesional skin (lines 95-97).

Line 195-197. It is stated that there is a loss of epidermal LC and keratinocyte function in the context of an IFN-rich pro-inflammatory environment, leading to photosensitivity in nonlesional lupus skin. The data shown in Figure 1 shows that upregulation of IFNs is significantly higher in lesional than nonlesional skin. It would be helpful to discuss the implications of the lesional data also.

Please see our response to Point #1.

It would be helpful to compare lesional and nonlesional in terms of the signatures shown in Figure 1 G and H. Is nonlesional different in terms of the IFN subtypes relative to lesional skin?

We have now added the signatures of IFN subtypes in both lesional and non-lesional skin (Figure 1F, H).

Langerhans cells were lower in lesional than nonlesional DLE, so not sure the parallel with early diseased mice B6.Sle1yaa mice and DLE is correct. Presumedly established DLE is more like the older mice who didn't show a difference in Langerhans cells once the disease was established. Would suggest revising this section a bit.

We agree and no longer draw parallels between the reduced LC signature in B6.Sle1yaa and progression of disease. Please see our response to Essential revisions Point #2, fourth paragraph.

Reviewer #2 (Recommendations for the authors):Non-lesional skin in lupus model mice share an IFN-I signature and other functional gene expression modules with human lupus skin:1. Lesional and non-lesional single-cell RNA seq analysis was recently published by Bili et al. (https://doi.org/10.1126/scitranslmed.abn2263).

Thank you for your thoughtful review. We agree that this is a very relevant paper and we have made sure to reference it.

2. Important literature is missing: Keratinocytes in non-lesional lupus skin produce IFN-kappa (https://doi.org/10.1038/s41584-022-00826-z). Upregulation of interferon-stimulated genes (ISGs) was already described in the blood and non-lesional skin of ANA-positive individuals. This ISG upregulation was indeed lower compared to lesional skin from SLE patients (https://doi.org/10.1038/s41467-020-19918-z).

We have now added the two excellent references and expanded our description of literature documenting an IFN-rich environment in non-lesional human skin in the Introduction (lines 118-128).

3. The MRL/lpr lupus-prone model has a less type I IFN-mediated disease, so it is not surprising that the authors found less significant changes compared to the control model.

We had mistakenly shown lesional DLE data rather than non-lesional data for Figures 1A-D in our original submission. The non-lesional skin has a weaker IFN signature than lesional skin (see Figure 1D , Figure 1—figure supplement 1) and the difference in magnitude between human non-lesional skin and controls compared to the LPR model and control (compare Figure 1—figure supplement 1vs Figure 2—figure supplement 1A) is less stark than previous. We have corrected the error in Figure 1A-D and the associated text in this revision. Please also see our response to Essential Revisions point #2, second paragraph.

4. The results from the B6.Sle1yaa lupus-prone model are equally less convincing. No difference in LC numbers was reported, whilst the authors only found Cd207 expression downregulation in 5-month deceased mice. These results are hard to interpret and to be associated with pathology in humans.

In our revised manuscript, we have made a point of differentiating among the effects of IFN on LC numbers, LC ADAM17 protein expression, and LC ADAM17 sheddase activity and find that IFN-I only consistently reduces LC ADAM17 sheddase function (summarized in Table 1, lines 366-375, 487-501). We do comment on the differences in LC numbers between human data and mouse models (lines 502-518), but, based on the data, our focus is on an IFN-LC ADAM17 sheddase function axis. Please also see our response to Essential Revisions point #2, fourth paragraph.

5. The results of ISG upregulation in the imiquimod (IMQ) model have been described in the literature many times; it is well established that imiquimod can induce an interferon response via TLR7 activation and upregulation of IRF target genes such as Irf7, Irf8 and Stat1.

Transcriptomic analyses of unpainted and painted skin from the IMQ lupus model mice has not been previously examined. IMQ over the short term has been used to induce a psoriasis model, but IMQ painting over several weeks to induce an SLE model is a comparatively new model (Yokogaswa et al. Arthritis Rheumatol 2014 66:694). We induce the model by painting one ear with IMQ, and our focus is on the unpainted contralateral “non-lesional” skin that presumably reflects the effects of systemic disease. Note that we have replaced the limited qPCR data with RNAseq data (Figure 2F-I, Figure 3E-F) to gain a more complete picture of the IFN signature and gene expression in this model. We now show that the non-lesional skin shares key pathways with non-lesional skin of human DLE and other lupus models (Figure 4B), while the IMQ-painted skin surprisingly did not yield an IFN signature (Figure 3—figure supplement 1), potentially reflecting tissue damage or some other distinct biology that reflects the prolonged direct exposure to IMQ.

6. The authors report that Isg15 was upregulated in the non-lesional ears of the IMQ mice, but the interpretation of the results needs more careful consideration. Isg15 was shown to be upregulated and secreted by plasmablasts in lupus patients inducing antibody-independent inflammation (https://doi.org/10.4049/jimmunol.1600624).

We have replaced the qPCR data with RNAseq to gain a more complete picture of the IFN signature in non-lesional skin of the IMQ model (Figure 2F-I, Figure 3E-F). We have not examined the cellular source of the Isg15 mRNA in the ear and it will be interesting to examine plasmablasts and other potential sources in future studies.

7. The authors conclude that "These data are consistent with the idea that non-lesional skin is primed for photosensitive responses and that LCs sit within an IFN-rich environment in both human lupus and multiple murine models that may cause LC dysfunction". Non-lesional skin has been shown to contribute to IFN-mediated responses as cited above. However, the authors do not provide robust in vitro or in vivo confirmation about how this IFN response mediates distinct phenotypes of LCs.

In our revised manuscript, we have now (1) made a point of distinguishing between IFN regulation of LC ADAM17 sheddase activity vs LC ADAM17 cell surface expression and LC numbers in our description of Figures 5-6, and (2) added Table 1 summarizing the effect by IFN on LC ADAM17 sheddase activity vs LC ADAM17 cell surface expression and LC numbers described in Figures 5-6. IFN-I only consistently inhibits LC ADAM17 sheddase activity, pointing to an IFN-LC ADAM17 sheddase function axis. Please see also response to Essential Revisions Point #1, first paragraph.

IFN-I inhibits LC ADAM17 function8. The authors assessed LC ADAM17 function by quantifying UVR-induced cell surface TNFR receptor 1 (TNFR1) shedding as "previously described". They need to provide insights about their methodological approach and not just simply cite a previously published article.

We have now expanded on the extensive validation of the LC ADAM17 sheddase activity that we had previously performed in (Shipman et al., Sci Trans Med, 2018 10: eaap9527). We also moved data from Supplementary Materials to new Figure 5A showing that our assay examining UVR-treatment of a mixture of epidermal cells induced TNFR1 loss on LCs in an LC ADAM17-dependent manner. (Discussed in lines 298-306). Please see also response to Essential Revisions Point #1, second paragraph.

9. The authors present data in Figure 4 only for the WT mice and some human LCs. LCs do not seem to be isolated from skin biopsies of lupus patients, whilst IFN-kappa is used for murine LCs but IFN-β for human LCs. Why is there a different approach to choosing other type I IFNs?

We now add data treating murine LCs with IFNb (Figure 5I, J) and show the same reduction in LC ADAM17 sheddase activity as with IFNb treatment of human LCs and IFNk treatment of murine LCs.

IFNAR is important for LC ADAM17 dysfunction in multiple lupus models10. The authors present the data on TNFR1 MFI change in the three different lupus-prone mice but no differences were detected in the actual ADAM17 protein expression. These results are difficult to interpret. Have the authors considered knocking out the mice for ADAM17 and reassessing the function of LCs?

The sheddase function of ADAM17 and other enzymes can be dissociated from regulation of protein expression. In our revised manuscript, we have made a point of clearly distinguishing the effects of IFN on LC numbers, LC ADAM17 protein expression, and LC ADAM17 sheddase activity and find that IFN-I only consistently reduces LC ADAM17 sheddase activity. Please also see our response to Essential Revisions Point #1, first paragraph.

Anti-IFNAR reduces photosensitivity in an EGFR and LC ADAM17-dependent manner11. The authors suggest that anti-IFNAR increased photosensitivity in non-lupus mice. How is this related to human data clearly showing that UVR is linked to photosensitivity by inducing IFN-kappa expression by keratinocytes both in vitro and in vivo (http://dx.doi.org/10.1136/annrheumdis-2018-213197 and https://doi.org/10.1038/s41467-020-19918-z)?

The IFN-I associated with disease is higher than normal, as demonstrated in these two references. These references also show that healthy epidermal cells do express IFN-I expression and will upregulate IFN-I upon stimulation. Normal skin upregulates IFN with injury and IFNAR has been shown to be needed for wound healing in normal mice, suggesting that IFN-I has a physiologic role in wound healing (Gregerio et al. 2010 *J Exp Med* 207:2921-2930)(Wolf et al. 2022 *JCI Insight* 7:e152765). Our interpretation of the anti-IFNAR-induced photosensitivity in non-lupus mice (Figure 7A-E) is that we disrupted the physiologic IFN needed in healthy skin. We have added a paragraph on this topic to the Discussion (lines 460-477).

IFN-I inhibits UVR-induced LC ROS expression12. In figure 7, the authors suggest that LC ROS is reduced in lupus model mice and restored by anti-IFNAR treatment. They use ROS indicator but they have not tried to measure if there is an increase in oxidative phosphorylation (OXPHOS) in the cells triggered by UVR. LCs from actual lupus patients are not used in the experiment.

We agree that our original Figure 7 did not differentiate between cytoplasmic vs mitochondrial sources of ROS, and increased OXPHOS would suggest increased mitochondria ROS. We now add Figure 8B-C using cytoplasmic and mitochondrial ROS indicators and show that UVR stimulates an early increase in cytoplasmic but not mitochondrial ROS. This is consistent with the critical role of cytoplasmic catalase in the early UVR-induced ROS generation found in keratinocytes (Heck et al. JBC 2003. 278:22432).

Reviewer #3 (Recommendations for the authors):– The analysis of the publicly available human data is extensive and largely confirms the already published findings of high IFN signatures in non-lesional and lesional DLE skin. While it is encouraging that the overall analyses of IFN-I response and immune cell signatures are consistent with already published findings, the sheer amount of data distracts from the finding most relevant to this study: decreased Langerhans Cell signature in non-lesional and lesional skin. The reviewer suggests the data in Figure 1 be summarized in 1 or 2 more comprehensive panels (e.g. a heatmap of cell signatures or cytokine signatures).

Thank you for your thoughtful feedback. We have summarized the GSVA findings in human DLE and murine SLE models as a heat map in Figure 4B.

– Similar to the human data, the findings of differential gene signatures in the skin of the murine lupus models could be summarized in a more comprehensive manner. The amount of data distracts from the most significant findings, particularly in B6.Sle1yaa model: decreased DC signature and increased IFN signature. Likewise, the reader is left to wonder at the end of Figure 2 if the IMQ-induced IFN response is accompanied by a decrease in LC numbers and/or function.

Thank you for the suggestion. We have summarized all the non-lesional skin GSVA findings in Figure 4B. The LC signature in IMQ mice is now addressed in the new RNAseq analysis (Figure 2F-I, Figure 3E-F, Figure 2—figure supplement 1B, Figure 3—figure supplement 1). LC counts in this model and response to anti-IFNAR are shown in Figure 6I and Figure 7L. We have also extensively revised the text describing the transcriptomic analyses of Figures 1-4 with the goal of being more clear.

– Summarizing the data as the IFN-I signature may be useful in Figure 2J to provide a better insight into the IFN-I response overall.

We have now replaced the PCR analysis with RNAseq showing IFN signatures in the IMQ model (Figure 2F-I, Figure 3E-F, Figure 2—figure supplement 1B, Figure 3—figure supplement 1).

– Computational analyses in Figures 1 and 2 emphasize the co-occurrence of a high IFN-I signature and a low LC and/or DC signature. It is not clear if the downregulation of the DC gene set indicates diminished presence of LCs in the non-lesional skin of the lupus mouse models or "reflects decreased LC function" as the authors suggest.

We now do the GSVA in Figures 1, 3,4, and Figure 3—figure supplement 1 using an LC signature rather than a more general dendritic cell signature. While the reduced LC signature in human DLE correspond to reduced LC numbers in direct tissue analysis, the reduced LC gene signature in B6.Sle1yaa non-lesional skin in the 5 month old mice examined in our current report seems at odds with the unchanged LC numbers in the 10 month old mice we examined in (Shipman et al., SciTransMed 2018 ,10(454):eaap9527). Without doing direct cellular phenotyping and further quantification, we don’t know the extent to which the reduced LC signatures reflect reduced LC numbers at 5 months of age or reduced expression of markers in the LCs. In this extensively revised manuscript, however, our data clearly establish that IFN-I consistently regulates LC ADAM17 sheddase activity rather than LC ADAM17 expression or LC numbers. These data point to an IFN-LC ADAM17 sheddase function axis in skin. Please see also our responses to Essential Revisions Point #1 first paragraph and Point #2 fourth paragraph.

– The significance of the Th1 signature in the CLE cohort discussed in line #149 to the interferon signature is unclear.

Our emphasis in this report is on the non-lesional skin; this sentence referred to the original (published) analysis of lesional skin from the same cohort. We now add Figure 1F showing that the IFN signature in DLE lesional skin includes both IFN-I and IFNgamma signatures. We have added to the text “the IFNγ signature found in our analysis is consistent with the Th1 signature identified in the original study examining these gene expression data along with cellular phenotyping (Jabbari et al., 2014).” (Text lines 193-197).

– Unclear what figure the findings in lines #175-178 refer to?

The text describing the transcriptomic analyses of Figures 1-4 has been extensively revised, and the sentence in question has been edited out.

– Unclear what figure the findings in lines #310-315 refer to?

The text describing Figures 1-4 have been extensively revised, and the unclear portion of these sentences have been edited out.

– The specificity of ADAM17 for TNFR1 is not explicitly stated. The previously published data using the LCad17 mouse should be emphasized and some of the supplemental data included in the main figures.

We have now expanded on the extensive validation of the LC ADAM17 sheddase activity that we had previously performed in (Shipman et al., Sci Trans Med, 2018 10: eaap9527). We also moved data from Supplementary Materials to new Figure 5A showing that our assay examining UVR-treatment of a mixture of epidermal cells induced TNFR1 loss on LCs in an LC ADAM17-dependent manner. (Discussed in lines 298-306). Please see also response to Essential revisions Point #1, second paragraph.

– In Figure 7 it is not clear whether and how the ROS are being specifically measured in LCs.

We have expanded on our Methods describing the original and new ROS experiments (lines 662-678) in what is now Figure 8.

– Given the hypothesis that IFN-I may be the cause of a decreased DC signature in the mouse skin, it would be relevant to ask if this signature is also decreased in the IMQ model, which is a known model of IFN-induction as confirmed by the authors. Likewise, asking how anti-IFNAR treatment affects the DC signature / LC numbers would be important, in the absence and presence of UV. The authors indicate in Figure 5I that IMQ reduces LC numbers.

We have now performed RNAseq on skin from IMQ mice and also perform GSVA with a Langerhans cell gene set rather than a more general dendritic cell gene set. There is not a relationship between the IFN signature and LC signature. IFN signature is upregulated in the “non-lesional” skin from the unpainted contralateral ear but the LC gene signature is not altered (Figure 2F-I, Figure 3E-F). In the IMQ painted ipsilateral ear , there was a reduced LC signature but no upregulation of an IFN signature (Figure 3-supplement 1). In Figure 6I, we have added more n and now no longer see any change in LC numbers in the “non-lesional” unpainted contralateral ear. We do note that both our facs data and the gene expression data shown that the “non-lesional” skin shows a high level of variability with regard to LC numbers/signatures, similar to our observations in lupus nephritis patients, where a proportion of but not all patients showed reduced LC numbers (Shipman et al., Sci Trans Med, 2018 10: eaap9527). We have added data on the effects of anti-IFNAR on IMQ LC counts both in the absence of UVR (Figure 6I) and after in vivo UVR exposure (Figure 7L); anti-IFNAR does not affect LC numbers in either condition. We have also added a paragraph addressing LC numbers humans and mouse models in the Discussion (lines 502-518). Note that in this extensively revised manuscript, we show that IFN-I consistently regulates LC ADAM17 sheddase activity rather than LC ADAM17 expression or LC numbers, thus pointing to an IFN-LC ADAM17 sheddase function axis in skin (discussed at length in response to Essential Revisions Point #1 first paragraph).

– Decreased inflammation in LCad17 mice in the IMQ+UV model is unexpected. Previous studies by this group showed increased UV-induced inflammation in the absence of LC-ADAM17 (Shipman et al. 2018). Is it possible that IMQ treatment prior to UV sets off a negative feedback loop that counteracts absence of Adam17? If the authors treated LCad17 mice with anti-IFNAR IgG prior to UV in the absence of IMQ, would the results defer? UV light is an inducer of IFN itself so may serve as the relevant stimulus.

We had shown that IMQ-treated LCAd17 mice had the same number of monocytes in UVR-treated skin as WT IMQ mice but , in normalized data, reduced neutrophils. Our main comparison was not between WT and LCAd17 cell numbers and when we went back to the primary data; we found that while we had multiple experiments comparing WT with WT+anti-IFNAR and LCAd17 with LCAd17+anti-IFNAR, we had too few experiments that included WT and LCAd17 together to be able to compare them to make firm conclusions. Here, we have put the neutrophil data showing absolute numbers into Figure 7K to be parallel to the graph showing absolute numbers of monocytes and removed the comparison between WT and LCAd17 mice.

– UV light is an important inducer of IFN. Authors have previously shown that UV also induces Adam17 expression. Therefore, the question remains whether a high baseline IFN signature in lupus skin suppresses UV-induced Adam17 expression?

Focusing on UVR-induced LC ADAM17 sheddase activity ---yes, our data support that the high baseline IFN in lupus skin suppresses UVR-induced ADAM17 sheddase function. We show that any modest modulation of ADAM17 protein expression that we observe in our different systems do not correlate with modulation of ADAM17 activity, and that IFN only consistently inhibited LC ADAM17 sheddase activity (new Table 1). We did show that murine LCs express *Adam17* mRNA in (Shipman et al., Sci Trans Med, 2018 10: eaap9527) , but the trend toward an increase with UVR was not denoted as a significant effect; this lack of *Adam17* mRNA change is consistent with what we also see in unpublished RNAseq data. Regarding UVR-induced IFN and how that might modulate LC ADAM17 sheddase function—the enzymatic activity is induced within 45 minutes of UVR exposure, based on our LC ADAM17 sheddase assays, whereas UVR-induced IFN seems to take hours (Skopelja-Gardner et al., Sci Rep, 2020 10:7908); it would be interesting to see if that later-onset induced IFN functions in part to turn off ADAM17 sheddase activity in healthy hosts.

– A direct mechanistic link between high IFN-I and loss of Adam17 activity driving photosensitive reactions could be strengthened. Would blocking Adam17 with a blocking antibody suppress photosensitive reactions in lupus mouse models? Would treating LCAd17 mice with IFN fail to enhance or diminish UV-induced inflammation?

We showed in (Shipman et al., Sci Trans Med, 2018 10: eaap9527) that LCAd17 mice lacking ADAM17 in LCs are photosensitive and that compensating for the loss of LC ADAM17 sheddase activity in LPR mice by topical EGFR ligand reduced photosensitivity. These data supported the idea that the reduction in LC ADAM17 sheddase activity contributed to photosensitivity in lupus model mice. In this manuscript, we address the mechanistic link between IFN and ADAM17 function in photosensitivity by showing that anti-IFNAR reduces UVR-induced skin inflammation in an EGFR-dependent manner in LPR mice (Figure 7G-I) and in an LC ADAM17-dependent manner in IMQ mice (Figure 7J-L). For (Figure 7J-L), we induced the IMQ model in WT and LCAd17 mice. While this latter experiment by itself does not differentiate between LC ADAM17 sheddase function and expression levels, we have now clarified that IFN-I regulates LC ADAM17 sheddase function, and this is uncoupled from LC ADAM17 expression (Table 1), further supporting the idea that anti-IFNAR is reducing UVR-induced skin inflammation in an LC ADAM17 function-dependent manner.

[Editors’ note: what follows is the authors’ response to the second round of review.]

The manuscript has been improved but there are some remaining issues that need to be addressed, as outlined below. Please respond to this as quickly as possible; it should not be necessary to consult reviewers again for the final revision.Reviewer 2 remarks that IFN enrichment seems too high for all clusters. Can you please more fully explain your conclusion regarding the IFN signature?Reviewer #2 (Recommendations for the authors):The authors have made a significant effort to address the comments pointed in the first review by reanalysing data, introducing new results, and updating essential literature missing.However, I would like to express some concerns about to original validity of the data. The authors state in the revised manuscript that "We had mistakenly shown lesional DLE data rather than non-lesional data for Figures 1A-D in our original submission". It would be impossible for the reviewing process to validate all data presented and a good will is expected. The authors should be accountable for presenting the right data and results in the first place.The authors also state that "they have replaced the limited qPCR data with RNAseq data (Figure 2F-I, Figure 3E-F) to gain a more complete picture of the IFN signature and gene expression in this model". From the data presenting in Figure 4, I can see how "the non-lesional skin shares key pathways with non-lesional skin of human DLE and other lupus models (Figure 4B), while the IMQ-painted skin surprisingly did not yield an IFN signature (Figure 3—figure supplement 1), potentially reflecting tissue damage or some other distinct biology that reflects the prolonged direct exposure to IMQ". IFN-enrichment appears to be high in all clusters.I would strongly recommend that the revised manuscript requires an expert bioinformatic review to analyse the computational approaches and reported results.

Figure 3 – supplement 1 shows that ipsilateral/painted ear in the IMQ model does not have an enrichment of the overall IFN gene set. In our manuscript, we had noted this lack of enrichment, and Reviewer 2 is noting that the ipsilateral/painted ear does, however, have enrichment of in two of the gene sets used to understand responses to specific IFNs -- the IFNB1 and IFNW1 gene sets. We have now amended our text to point out the enrichment of the IFNB1 and IFNW1 gene sets despite the lack of enrichment of the overall IFN gene set (lines 270-273).